# Leveraging Diffusion Model as Pseudo-Anomalous Graph Generator for Graph-Level Anomaly Detection

**Jinyu Cai** [1]  **Yunhe Zhang** [2] [*]  **Fusheng Liu** [1]  **See-Kiong Ng** [1]

## Abstract

A fundamental challenge in graph-level anomaly detection (GLAD) is the scarcity of anomalous graph data, as the training dataset typically contains only normal graphs or very few anomalies. This imbalance hinders the development of robust detection models. In this paper, we propose **A**nomalous **G**raph **Diff**usion (AGDiff), a framework that explores the potential of diffusion models in generating pseudo-anomalous graphs for GLAD. Unlike existing diffusion-based methods that focus on modeling data normality, AGDiff leverages the latent diffusion framework to incorporate subtle perturbations into graph representations, thereby generating pseudo-anomalous graphs that closely resemble normal ones. By jointly training a classifier to distinguish these generated graph anomalies from normal graphs, AGDiff learns more discriminative decision boundaries. The shift from solely modeling normality to explicitly generating and learning from pseudo graph anomalies enables AGDiff to effectively identify complex anomalous patterns that other approaches might overlook. Comprehensive experimental results demonstrate that the proposed AGDiff significantly outperforms several state-of-the-art GLAD baselines.

## 1. Introduction

Graph-level anomaly detection (GLAD) (Akoglu et al., 2015; Qiao et al., 2024a; Liu et al., 2024b; Cai et al., 2024d) focuses on the fundamental challenge of identifying irregularities in graph-level data, where anomalous graphs significantly deviate from the normal graph distribution. GLAD is crucial across numerous domains, from detecting abnormal patterns in social networks (Yu et al., 2016; Zhang et al., 2025; Qiao & Pang, 2023) to identifying anomalous proteins in biological systems (Li et al., 2022). Unlike node- or edge-level anomaly detection (Duan et al., 2023; Qiao et al., 2024a;b; Pan et al., 2025), GLAD poses unique challenges as it requires modeling complex topological and geometric structures at the entire graph level.

The evolution of GLAD methods has witnessed several key developments. Classical graph kernel methods, such as the Weisfeiler-Lehman kernel (Shervashidze et al., 2011), random walk kernel (Vishwanathan et al., 2010), have established a foundation for GLAD by computing graph similarity matrices based on structural features. These approaches excel at capturing local topological patterns, but typically struggle with computational complexity. Recent deep learning based GLAD methods have emerged in both unsupervised and semi-supervised paradigms. Unsupervised methods typically employ graph neural networks (GNNs) (Kipf & Welling, 2017; Xu et al., 2019; Huang et al., 2023; Wan et al., 2024; Tu et al., 2025) to learn graph-level features for anomaly detection, with techniques such as one-class classification (Ruff et al., 2018; Qiu et al., 2022; Zhang et al., 2024), information bottleneck (Liu et al., 2023a), knowledge distillation (Ma et al., 2022), and graph reconstruction (Kim et al., 2024) to learn normality patterns without labeled anomalies. Semi-supervised approaches (Zhang et al., 2022; Xu et al., 2024) generally leverage limited graph anomalies to train a classifier as the anomaly detector. Even though only a small fraction of labeled anomalies are available, semi-supervised approaches have demonstrated remarkable performance improvement in detecting graph anomalies.

Despite these advancements, several critical limitations persist in existing approaches. Unsupervised methods focus on modeling normal graph distributions, they generally struggle to distinguish intricate or subtle anomalies, especially those near the boundaries of normal graphs, due to the lack of explicit supervised information. On the other hand, semi-supervised approaches can leverage limited labeled anomalies to enhance decision boundary learning. However, their effectiveness is constrained by the scarcity and diversity of labeled anomalous graphs, which limits their

[1]Institute of Data Science, National University of Singapore, Singapore [2]Department of Computer and Information Science, SKL-IOTSC, University of Macau, Macau, China. Correspondence to: Yunhe Zhang <zhangyhannie@gmail.com>.

*Proceedings of the $42^{nd}$ International Conference on Machine Learning*, Vancouver, Canada. PMLR 267, 2025. Copyright 2025 by the author(s).

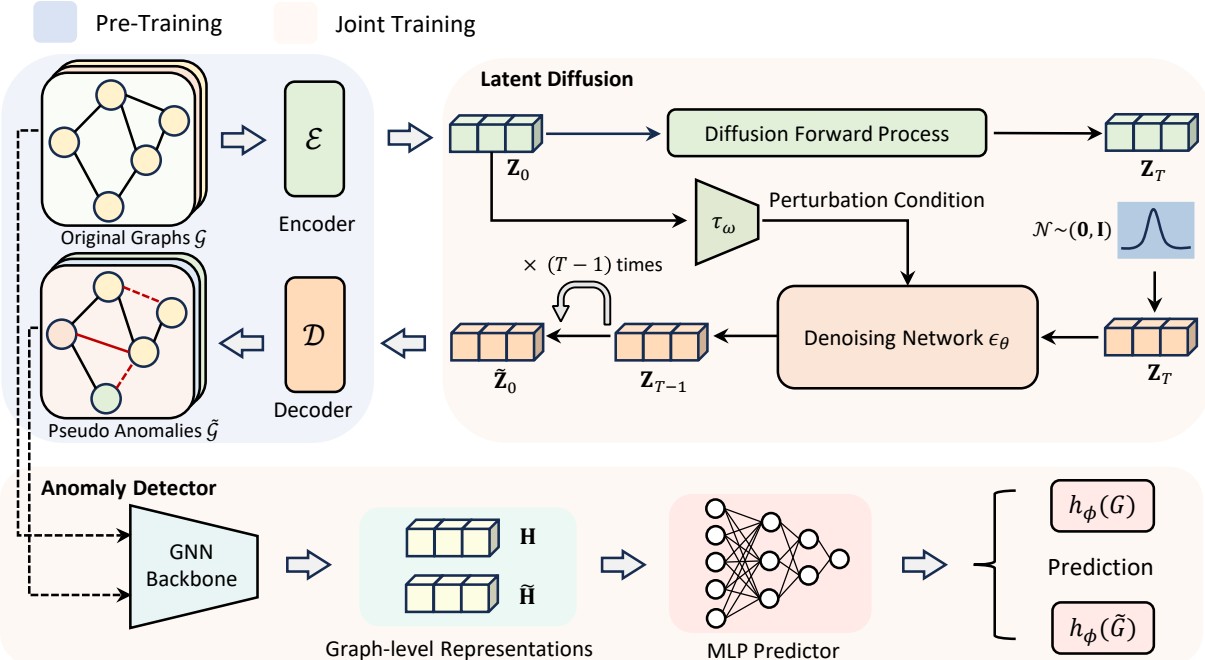

*Figure 1.* An illustration of the proposed AGDiff framework. The framework consists of three main components: (1) Pre-training, (2) Latent diffusion-based pseudo-anomalous graph generation, and (3) Anomaly detector. The pre-training phase learns a structured latent space via a reconstruction model. The latent diffusion process generates pseudo-anomalous graphs by perturbing latent embeddings through a forward diffusion process and a reverse denoising process. Finally, the anomaly detector distinguishes between normal and generated pseudo-anomalous graphs through joint training.

generalizability to rare or unseen anomaly types.

To address these challenges, we propose Anomalous Graph Diffusion (AGDiff), a novel framework (see Figure 1) that leverages the generative capabilities of diffusion models to generate diverse pseudo-anomalous graphs for GLAD. By introducing a conditioned latent diffusion process within a well-trained graph representation learning model, AGDiff ensures the preservation of essential graph properties while imposing controllable perturbations in the latent space. Additionally, AGDiff employs a joint training paradigm that simultaneously optimizes pseudo-anomalous graph generation and anomaly detection. Our algorithm analysis (in **Appendix A**) highlights the advantages of AGDiff over traditional reconstruction-based methods by demonstrating that leveraging diverse pseudo-anomalous graphs enables a more robust and refined decision boundary for anomaly detection. By bridging the gap between generative modeling and discriminative learning, AGDiff offers a generalizable solution to the fundamental challenges of GLAD.

The primary contributions of this work are as follows:

1. We propose AGDiff, the first framework that explores the potential of diffusion models to mitigate the anomaly scarcity challenge in GLAD.

2. We propose a latent diffusion process with perturbation conditions to generate pseudo-anomalous graphs without relying on any labeled anomalies for improving decision boundary learning.

3. We demonstrate the superiority of AGDiff across extensive comparisons with state-of-the-art GLAD baselines on diverse graph benchmarks.

## 2. Related Work

### 2.1. Graph-level Anomaly Detection

Graph-level anomaly detection (GLAD) (Akoglu et al., 2015; Qiao et al., 2024a; Cai et al., 2024b) aims to identify graphs whose structural or attribute patterns deviate from the majority, with widespread real-world applications in fraud detection, bioinformatics, and cybersecurity. Traditional approaches, such as graph kernels (Borgwardt & Kriegel, 2005; Vishwanathan et al., 2010; Shervashidze et al., 2011), struggle to model the intricate dependencies within graph data due to their reliance on hand-crafted features. In recent years, the emergence of GNNs (Kipf & Welling, 2017; Xu et al., 2019; Li et al., 2023; Liu et al., 2023b; Cai et al., 2024a;c) has substantially advanced GLAD via their expressive graph representation learning capabilities. Latest

GLAD approaches have explored knowledge distillation for global-local anomaly identification (Ma et al., 2022), graph transformation learning to mitigate representation collapse (Qiu et al., 2022), counterfactual graph generation for robust detection (Xiao et al., 2024a), spectral analysis to capture global graph anomaly properties (Dong et al., 2024), and explainable learning for interpretable anomaly detection (Liu et al., 2023a), etc. Additionally, semi-supervised learning (Zhang et al., 2022) is also studied to alleviate the data imbalance problem in GLAD.

Despite these advancements, existing GLAD methods still face significant limitations. For example, the reconstruction flip phenomenon identified by Kim et al. (2024) reveals that the common assumption in existing approaches, *i.e.*, anomalies necessarily exhibit higher reconstruction errors, is not universally applicable. Moreover, existing approaches are trained with only normal graph data or just a small set of anomalies, which limits their generalizability to unseen anomalies. Consequently, in the proposed AGDiff framework, we aim to adaptively generate diverse pseudo-anomalous graphs to enhance the capacity to detect subtle and complex anomalies while mitigating data imbalance issues. Furthermore, the generative process inherently improves interpretability by revealing how pseudo-anomalous graphs are generated through controlled perturbations. Compared to other approaches, our method delivers a robust and interpretable solution for GLAD problems.

### 2.2. Diffusion Model

Diffusion models (Ho et al., 2020; Song et al., 2021; Yang et al., 2023) have emerged as a new paradigm for generative modeling, wherein data are gradually corrupted through a diffusion forward process and subsequently denoised through a learned reverse process. Early work by Ho et al. (Ho et al., 2020) demonstrated that diffusion processes could capture high-dimensional densities more effectively than traditional architectures such as GANs (Goodfellow et al., 2020; Wang et al., 2021; Cai et al., 2024e) and VAEs (Kingma, 2013). In the context of image anomaly detection, diffusion-based methods have demonstrated impressive capabilities in domains such as industrial defect inspection (Zhang et al., 2023; Tebbe & Tayyub, 2024) and medical anomaly detection (Wolleb et al., 2022; Bercea et al., 2024), as they excel by modeling intricate pixel-level dependencies, thereby enabling the identification of subtle deviations through the pixel-level reconstruction of images. Recent advances have investigated the diffusion frameworks for graph-structured data, demonstrating notable potential for tasks such as graph generation (Kong et al., 2023), node classification (Yang et al., 2024), and graph anomaly detection (Li et al., 2024; Liu et al., 2024a; Xiao et al., 2024b).

However, existing diffusion-based anomaly detection meth-ods (Wolleb et al., 2022; Tebbe & Tayyub, 2024; Li et al., 2024) primarily focus on modeling normality and expect anomalies to emerge naturally as reconstruction outliers, presupposing that anomalies deviate significantly in topology or node features. Such an assumption falters when anomalies are inherently subtle, especially in graphs where localized irregularities can be easily "diluted" by global reconstruction. Different from the existing approaches, we propose to leverage diffusion models as a pseudo-anomalous graph generator through controlled perturbations introduced during the diffusion process. We expect the generated pseudo graphs to resemble the normal graphs, so that they can provide explicit supervision for learning a robust anomaly detection model.

## 3. Preliminary

The diffusion model, *e.g.*, denoising diffusion probabilistic model (DDPM) (Ho et al., 2020), is one of the generative model families, which typically contains a forward diffusion process that perturbs the original data $\mathbf{x}_0$ into a noisy sample $\mathbf{x}_t$ by progressively adding Gaussian noise over $T$ time steps, and a reverse denoising process that attempts to recover $\mathbf{x}_0$ from $\mathbf{x}_t$ by removing the noise step-by-step. Specifically, the forward diffusion process is defined as:

$$\mathbf{x}_t = \sqrt{\bar{\alpha}_t}\mathbf{x}_0 + \sqrt{1 - \bar{\alpha}_t}\epsilon_t, \quad \epsilon_t \sim \mathcal{N}(\mathbf{0}, \mathbf{I}). \quad (1)$$

Here, $\bar{\alpha}_t = \prod_{i=1}^{t} \alpha_i = \prod_{i=1}^{t}(1 - \beta_i)$, where $\beta_i$ denotes the noise variance schedule that imposed at each time step. The reverse denoising process can be understood as a series of learned denoising steps to recover the original data $\mathbf{x}_0$ from the noise sample $\mathbf{x}_t$. At each time step $t$, $\mathbf{x}_{t-1}$ is reconstructed by:

$$\mathbf{x}_{t-1} = \frac{1}{\sqrt{\alpha}} \left( \mathbf{x}_t - \frac{1 - \alpha_t}{\sqrt{1 - \bar{\alpha}_t}} \epsilon_\theta(\mathbf{x}_t, t) \right) + \bar{\beta}\mathbf{z}, \quad (2)$$

where $\epsilon_\theta(\mathbf{x}_t, t)$ is a learnable noise predict function, $\bar{\beta} = \frac{1 - \bar{\alpha}_{t-1}}{\bar{\alpha}_t}$, and $\mathbf{z} \sim \mathcal{N}(\mathbf{0}, \mathbf{I})$ is sampled from a normal Gaussian distribution. The training of diffusion models revolves around learning the reverse process that can accurately recover data from the noisy representation, where the objective can be expressed as:

$$\min_\theta \ \mathbb{E}_{t \sim [1,T], \mathbf{x}_0 \sim q(\mathbf{x}_0), \epsilon \sim \mathcal{N}(\mathbf{0}, \mathbf{I})} \left[ \| \epsilon - \epsilon_\theta(\mathbf{x}_t, t) \|_2^2 \right]. \quad (3)$$

After training the model, the original data $\mathbf{x}_0$ can be reconstructed by iteratively removing predicted noise $\epsilon_\theta(\mathbf{x}_t, t)$ from $\mathbf{x}_t$.

## 4. Methodology

### 4.1. Problem Formulation

Let $\mathcal{G} = (\mathcal{V}, \mathcal{E})$ denote a graph where $\mathcal{V}$ represents the set of vertices and $\mathcal{E} \subseteq \mathcal{V} \times \mathcal{V}$ represents the set of edges.

Each graph $\mathcal{G}$ is characterized by its adjacency matrix $\mathbf{A} \in \{0,1\}^{n \times n}$ and node feature matrix $\mathbf{X} \in \mathbb{R}^{n \times d}$, where $n = |\mathcal{V}|$ is the number of nodes and $d$ is the dimension of node features. In an unsupervised GLAD problem (Akoglu et al., 2015; Qiao et al., 2024a), given a normal graph set $\mathcal{G} = \{G_1, G_2, \ldots, G_N\}$ is given by sampling from an unknown normal graph distribution $\mathcal{P}_{\text{normal}}$. Our goal is to learn an anomaly detection function $f : G \to \mathbb{R}$ that identifies graphs deviating from $\mathcal{P}_{\text{normal}}$. Specifically, for a test graph $G_{\text{test}}$, we aim to estimate:

$$f(G_{\text{test}}) = P(G_{\text{test}} \notin \mathcal{P}_{\text{normal}} | G_{\text{test}}). \tag{4}$$

While some semi-supervised methods (Zhang et al., 2022) attempt to introduce a small fraction of real anomalies into training a classifier as the anomaly detector, the scarcity and limited diversity of anomalies make it challenging to generalize to unseen anomalous patterns. Therefore, we are curious whether the normality induced by normal graph distribution can be leveraged as a special kind of "supervised information" to facilitate the training of a more robust graph anomaly detector. In this work, we aim to generate a set of pseudo-anomaly graphs by introducing controlled perturbations for normal graphs under a latent diffusion framework. Then, we jointly train a graph anomaly detector with the generation framework so that the decision boundary learning and the pseudo-anomalous graph generation can be mutually refined. We will describe the details of our framework in the following sections.

### 4.2. Modeling Normality via Variational Inference

In the context of graph-level anomaly detection, one of the foundational challenges is the modeling of the normal graph distribution, as anomalies are typically defined as deviations from the normality learned from a given normal graph set $\mathcal{G}_{\text{normal}}$. To achieve this, we first pre-train a graph representation learning model aiming at capturing the normality of graphs. Particularly, we leverage variational inference to learn a probabilistic mapping from graph space to continuous latent space, as it naturally provides a well-regularized manifold for subsequent perturbation-based anomaly generation. For a given graph set $\mathcal{G} = \{\mathbf{A}, \mathbf{X}\}$, we approximate the posterior distribution over the latent representation $\mathbf{Z} \in \mathbb{R}^{n \times d_z}$ as:

$$q(\mathbf{Z}|\mathbf{X}, \mathbf{A}) = \prod_{i=1}^{n} q(\mathbf{z}_i|\mathbf{X}, \mathbf{A}),$$
$$\text{w.r.t } q(\mathbf{z}_i|\mathbf{X}, \mathbf{A}) = \mathcal{N}(\mathbf{z}_i|\boldsymbol{\mu}_i, \text{diag}(\boldsymbol{\sigma}_i^2)), \tag{5}$$

where $\boldsymbol{\mu}$ and $\boldsymbol{\sigma}$ are parameterized by $\boldsymbol{\mu} = \text{GNN}_{\boldsymbol{\mu}}(\mathbf{X}, \mathbf{A})$ and $\log \boldsymbol{\sigma} = \text{GNN}_{\boldsymbol{\sigma}}(\mathbf{X}, \mathbf{A})$, respectively. By using the reparameterization trick to enable gradient propagation, we sample $\mathbf{Z} = \boldsymbol{\mu} + \boldsymbol{\sigma} \odot \tau$, where $\tau \sim \mathcal{N}(\mathbf{0}, \mathbf{I})$. The reconstruction process involves recovering both structural (denoted by

$\hat{\mathbf{A}}$) and attribute (denoted by $\hat{\mathbf{X}}$) information for graphs:

$$\hat{\mathbf{A}} = \mathcal{T}(\mathbf{Z}\mathbf{Z}^{\top}), \quad \hat{\mathbf{X}} = \mathcal{D}(\mathbf{Z}), \tag{6}$$

where $\mathcal{T}(\cdot)$ denotes a `Sigmoid` transformation function, and $\mathcal{D}(\cdot)$ represents an MLP-based decoder for attribute reconstruction. The model is optimized by minimizing:

$$\begin{aligned} \mathcal{L}_{\text{pretrain}} =& \ell_r^{\text{attr}} + \ell_r^{\text{edge}} + \ell_{\text{KL}} \\ =& \sum_{i=1}^{N} (\|\mathbf{X}_i - \hat{\mathbf{X}}_i\|_F^2 + \mathcal{H}(\mathbf{A}_i, \hat{\mathbf{A}}_i) \\ & - \text{KL}(q(\mathbf{Z}_i|\mathbf{X}_i, \mathbf{A}_i)|\mathcal{P}(\mathbf{Z}))), \end{aligned} \tag{7}$$

where $\mathcal{H}(\cdot)$ denotes the binary cross-entropy loss for edge reconstruction, and the KL divergence term regularizes the learned distribution towards a prior $\mathcal{P}(\mathbf{Z}) = \prod_i \mathcal{N}(\mathbf{z}_i|\mathbf{0}, \mathbf{I})$, which encourages a smooth and well-structured latent space and prevents the model from overfitting to a narrow or degenerate latent manifold (Kipf & Welling, 2016).

### 4.3. Generating Anomalous Graphs via Latent Diffusion

Building on a well-structured latent space that effectively captures normal graph patterns, we propose a novel approach that utilizes latent diffusion models to generate pseudo-anomalous graphs. This approach represents a substantial departure from conventional diffusion-based anomaly detection methods, which typically aim to model the normality of data via diffusion models. Instead, we exploit the generative power of diffusion models to generate diverse pseudo-anomalous graphs by introducing controlled perturbations during the latent diffusion process. The smooth and continuous geometry of the latent space provides a foundation for this process, enabling the preservation of critical structural properties and meaningful deviations from normality. Our key insights are: (1) the continuous nature of the latent space makes it particularly suitable for the diffusion process, and (2) the asymptotic nature of the diffusion process allows fine-grained control over the deviation from normality.

Given a normal graph set $\mathcal{G} = \{\mathbf{X}, \mathbf{A}\}$, we first obtain its latent representation $\mathbf{Z} = \mathcal{E}(\mathbf{X}, \mathbf{A})$ through the pre-trained model. To generate controlled perturbations, we design a conditional latent diffusion process that consists of two key components: a forward process that gradually adds noise to the latent representation, and a reverse process that learns to denoise while preserving essential graph properties. The forward process progressively corrupts the latent representation by injecting Gaussian noise over $T$ time steps:

$$\mathbf{z}_t = \sqrt{\bar{\alpha}_t} \mathbf{z}_0 + \sqrt{1 - \bar{\alpha}_t} \epsilon_t, \quad \epsilon_t \sim \mathcal{N}(\mathbf{0}, \mathbf{I}), \tag{8}$$

where $\mathbf{z}_0$ is initialized by $\mathbf{Z}$, and $\bar{\alpha}_t = \prod_{i=1}^{t} \alpha_i = \prod_{i=1}^{t} (1 - \beta_i)$ determines the noise schedule through $\beta_i$.

This schedule is crucial as it determines the degree of perturbation at each step.

A key challenge in generating pseudo-anomalous graphs is maintaining the balance between structural preservation and anomaly injection. To address this, we design a conditional reverse process:

$$\mathbf{z}_{t-1} = \frac{1}{\sqrt{\alpha}} \left( \mathbf{z}_t - \frac{1-\alpha_t}{\sqrt{1-\bar{\alpha}_t}} \epsilon_\theta(\mathbf{z}_t, t, \mathbf{c}) \right) + \tilde{\beta}\mathbf{v}, \quad (9)$$

where $\epsilon_\theta(\mathbf{z}_t, t, \mathbf{c})$ is our conditional noise prediction network, $\tilde{\beta} = \frac{1-\bar{\alpha}_{t-1}}{\bar{\alpha}_t}$, and $\mathbf{v} \sim \mathcal{N}(\mathbf{0}, \mathbf{I})$. The condition vector $\mathbf{c}$ is obtained via a perturbation condition model $\tau_\omega$ to add auxiliary noise information to the generation process:

$$\mathbf{c} = \tau_\omega(\mathbf{z}_0) = \sigma(\mathbf{W_c}(\mathbf{z}_0 + \eta) + \mathbf{b_c}), \quad (10)$$

where $\eta \sim \mathcal{N}(\mathbf{0}, \boldsymbol{I})$ is a Gaussian noise vector that introduces perturbations to the initial latent representation, and the learnable weight matrix $\mathbf{W_c}$ and bias vector $\mathbf{b_c}$ transform the perturbed representation to a more expressive feature space through a non-linear activation function $\sigma(\cdot)$. The condition vector $\mathbf{c}$ is concatenated with the latent variable $\mathbf{z}_t$ at each denoising step to introduce controlled variations during the diffusion process.

The condition vector $\mathbf{c}$ is crucial for guaranteeing the generation quality of pseudo graph anomalies. If we directly use the perturbed normal data (*e.g.*, via random noise $\eta$) as pseudo anomalies, it would constrain the diversity of anomalous patterns the model encounters, as such perturbations are static and lack adaptability during training. In contrast, we propose to perturb the initial latent embedding through a learnable perturbation transformation $\mathbf{c}$, which injects additional variability into the latent diffusion process. The perturbations are learnable and dynamically adjusted during joint training with the anomaly detector. As the anomaly detector improves, the perturbation mechanism evolves and generates increasingly sophisticated and diverse pseudo anomalies to refine the decision boundary. This ensures that the denoising network is conditioned to deviate from purely "normal" reconstructions and allows the model to be exposed to a broader spectrum of potential anomalies, thereby enhancing its robustness and generalization. The latent diffusion model is then optimized by minimizing:

$$\mathcal{L}_{\text{diff}} = \mathbb{E}_{\mathbf{z}_0, \epsilon, t, \mathbf{c}} \left[ \|\epsilon - \epsilon_\theta(\mathbf{z}_t, t, \mathbf{c})\|_2^2 \right]. \quad (11)$$

Through the conditional diffusion process, we can iteratively generate perturbed latent features $\tilde{\mathbf{z}}_0$, *i.e.*, $\mathbf{z}_t \rightarrow \mathbf{z}_{t-1} \rightarrow \cdots \rightarrow \tilde{\mathbf{z}}_0$ for each graph. Therefore, the pseudo-anomalous graphs can be obtained through the pre-trained decoder via: $\tilde{\mathbf{A}} = \mathcal{T}(\tilde{\mathbf{Z}}\tilde{\mathbf{Z}}^\top)$ and $\tilde{\mathbf{X}} = \mathcal{D}(\tilde{\mathbf{Z}})$, where $\tilde{\mathbf{Z}} = \{\tilde{\mathbf{z}}_0^{(i)}\}_{i=1}^N$.

### 4.4. Detecting Anomalies from Subtle Deviations

Conventional semi-supervised GLAD approaches generally struggle in scenarios where real anomalous samples are very

---

**Algorithm 1** AGDiff

**Input:** Input graph set $\mathcal{G}$, number of GIN layers $K$, learning rate $\rho$, diffusion time steps $T$, training epochs $Epochs$.

**Output:** The anomaly detection scores.

1: Initialize the network parameters;
2: Pre-train the representation learning model by minimizing Eq. (7);
3: Freeze the network parameters of the pre-trained model;
4: **for** $epoch = 1$ to $Epochs$ **do**
5:      Sample a mini-batch of graphs from graph dataset $\mathcal{G}$;
6:      Obtain latent embeddings $\mathbf{Z}$ from the pre-trained encoder;
7:      Perform forward diffusion process on $\mathbf{Z}$ via Eq. (8);
8:      Obtain the condition vector via Eq. (10);
9:      Generate pseudo-anomalous embeddings $\tilde{\mathbf{Z}}$ through the reverse diffusion process via Eq. (9);
10:      Decode $\tilde{\mathbf{Z}}$ via pre-trained decoder to generate pseudo-anomalous graphs set $\tilde{\mathcal{G}}$;
11:      Predict scores for the normal and generated pseudo-anomalous graphs via Eq. (12);
12:      Jointly update the parameters of the latent diffusion model and anomaly detector by minimizing Eq. (14);
13: **end for**
14: Compute anomaly detection scores for test graphs via the trained classifier $f_\theta$;
15: **Return:** The anomaly detection scores.

---

rare or even unavailable, as their reliance on labeled anomalies limits their capacity to generalize to unseen abnormal patterns. This challenge becomes particularly pronounced in real-world scenarios, where anomalies are inherently rare and diverse. To overcome this limitation, we propose a joint learning framework that leverages generated pseudo-anomalous graphs to facilitate the training of the anomaly detector. These pseudo-anomalous graphs serve as proxies for real anomalies, which enables the model to learn more robust, adaptive decision boundaries that capture subtle deviations from normality.

In practice, we employ a GIN-based anomaly detector $h_\phi(\cdot)$ to distinguish between normal graphs and pseudo-anomalous graphs, which is defined by:

$$h_\phi(G) = \text{MLP}(\text{GIN}(\mathbf{X}, \mathbf{A})), \quad (12)$$

where $h_\phi(\cdot)$ parameterized by $\phi$ is comprised of a GIN-based backbone network $\text{GIN}(\cdot)$ and an MLP-based projector $\text{MLP}(\cdot)$. We employ a following binary cross-entropy loss $\mathcal{L}_{\text{cls}}$ to train the anomaly detector:

$$\mathcal{L}_{\text{cls}} = - \frac{1}{|\mathcal{G} \cup \tilde{\mathcal{G}}|} \sum_{G \in \mathcal{G} \cup \tilde{\mathcal{G}}} (y_G \log h_\phi(G) \\ + (1 - y_G) \log(1 - h_\phi(G))), \quad (13)$$

where $\mathcal{G}$ and $\tilde{\mathcal{G}}$ denote the normal and pseudo-anomalous graph sets, respectively. Note that we set $y_G = 1, \forall G \in \mathcal{G}$, and $y_G = 0, \forall G \in \tilde{\mathcal{G}}$ to train the anomaly detector. In particular, we jointly train the latent diffusion model and the anomaly detector by minimizing:

$$\mathcal{L} = \mathcal{L}_{\text{cls}} + \lambda \mathcal{L}_{\text{diff}} \qquad (14)$$

where $\lambda$ is the hyper-parameter that controls the trade-off between two objectives. $\mathcal{L}_{\text{diff}}$ guides the generation of pseudo graph anomalies, and $\mathcal{L}_{\text{cls}}$ enhances the discriminative power of the anomaly detector. In the inference phase, we can obtain the anomaly score of each test graph sample via the trained anomaly detector. Algorithm 1 summarizes the training procedure of AGDiff. The joint learning framework offers several unique advantages:

1. The latent diffusion model learns to generate increasingly challenging pseudo-anomalies that explore the decision boundary of the anomaly detector.

2. The gradient of the detector directs the diffusion process toward generating more informative pseudo-anomalous samples.

3. The iterative refinement between generation and detection leads to a more robust anomaly detector.

### 4.5. Computational Complexity Analysis

For a dataset of $N$ graphs, each with an average of $m$ nodes (feature dimension $d$), $|E|$ edges, and latent dimension $d_z$, the AGDiff framework operates in three phases:

1. **Pre-training:** An $L$-layer GIN is employed as the backbone network in the pre-training, where the overall complexity is $\mathcal{O}(NL(|E|d + md^2))$ due to the message aggregation ($\mathcal{O}(|E|d)$) and feature transformation ($\mathcal{O}(md^2)$).

2. **Pseudo Anomaly Generation:** This phase involves the computation of conditional vector ($\mathcal{O}(Nd_z^2)$) and a $T$-step latent diffusion process ($\mathcal{O}(NTmd_z^2)$) across all graphs.

3. **Decoding and Anomaly Detection:** Decoding involves the computation of node attributes and adjacency matrices from latent embeddings, which results in time complexity of $\mathcal{O}(N|E|d_z)$ when we apply a negative sampling strategy in practice. The computational complexity of the anomaly detector is similar to the pre-training stage, *i.e.*, ($\mathcal{O}(NL(|E|d + md^2))$), due to their similar network structure.

Therefore, the overall computational complexity of AGDiff is approximately $\mathcal{O}(NL(|E|d + md^2) + N(Tm + 1)d_z^2 + N|E|d_z)$, which is comparable with other state-of-the-art

baselines such as SIGNET, MUSE, DO2HSC. In addition, potential optimizations such as using parallel computation further enhance efficiency.

## 5. Experiments

### 5.1. Experimental Setup

**Datasets.** We conduct experiments with two types of graph benchmarks, including

- *Moderate-Scale Datasets*: MUTAG, DD, COX2, and ER_MD. These datasets primarily consist of molecular graphs and bioinformatics networks, where nodes and edges represent molecular structures or protein interactions.

- *Large-Scale Imbalanced Datasets*: SW-620, MOLT-4, PC-3, and MCF-7. Each graph in these datasets represents a chemical compound, with labels indicating whether it exhibits anti-cancer activity. The datasets are highly imbalanced as active compounds form only a small fraction of the total samples, which makes them well-suited for evaluating the robustness of anomaly detection methods in real-world, unbalanced settings.

We describe the details about the datasets and their characteristics in **Appendix B**. Note that we follow the settings in existing works (Ma et al., 2022; Zhang et al., 2022; Liu et al., 2023a) to treat the minority class as the anomalous class to better align with real-world scenarios where anomalies often correspond to rare events or unusual patterns.

**Implementation Details.** Due to the length limitation of the paper, please refer to **Appendix C** for the implementation details of the experiment, including the data split, network architecture, hyper-parameter setting, baseline setting, and computing resources.

**Compared Baselines.** To evaluate the effectiveness of the proposed AGDiff method, we compared it with two types of GLAD baselines, including: (1) *Graph Kernel Methods*: Short-Path (SP) kernel (Borgwardt & Kriegel, 2005), Weisfeiler-Lehman (WL) kernel (Shervashidze et al., 2011), NH (Hido & Kashima, 2009), Random Walk (RW) kernel (Vishwanathan et al., 2010), and (2) *GNN-based GLAD Methods*: OCGIN (Zhao & Akoglu, 2023), OCGTL (Qiu et al., 2022), GLocalKD (Ma et al., 2022), iGAD (Zhang et al., 2022), SIGNET (Liu et al., 2023a), MUSE (Kim et al., 2024), DO2HSC (Zhang et al., 2024).

**Evaluation Metrics.** To evaluate the anomaly detection performance of each method, we utilize two widely adopted metrics: Area Under the Curve (AUC) and F1-Score. The experimental results are reported as the mean values and

*Table 1.* Average AUCs and F1-Scores with standard deviation (10 trials) on four small and moderated graph datasets. The best results are marked in **bold**, and "OM" denotes out-of-memory.

| Method | MUTAG | | DD | | COX2 | | ER_MD | |
|---|---|---|---|---|---|---|---|---|
| | AUC | F1-Score | AUC | F1-Score | AUC | F1-Score | AUC | F1-Score |
| SP (Borgwardt & Kriegel, 2005) | 67.52±0.00 | 60.00±0.00 | 82.73±0.00 | 76.09±0.00 | 54.08±0.00 | 49.32±0.00 | 40.92±0.00 | 37.74±0.00 |
| WL (Shervashidze et al., 2011) | 60.00±0.00 | 89.12±0.00 | 81.57±0.00 | 74.64±0.00 | 49.32±0.00 | 50.19±0.00 | 37.74±0.00 | 45.71±0.00 |
| NH (Hido & Kashima, 2009) | 79.97±0.40 | 76.00±0.00 | 81.61±0.32 | 73.91±0.65 | 61.41±0.82 | 56.44±1.03 | 51.55±2.00 | 50.19±0.92 |
| RW (Vishwanathan et al., 2010) | 86.98±0.00 | 83.33±0.00 | OM | OM | 52.43±0.00 | 30.00±0.00 | 78.94±0.00 | 65.96±0.00 |
| OCGIN (Zhao & Akoglu, 2023) | 74.66±1.68 | 62.95±0.00 | 66.59±4.44 | 56.12±0.00 | 59.64±5.78 | 47.95±0.00 | 47.63±3.59 | 50.94±1.89 |
| OCGTL (Qiu et al., 2022) | 87.04±1.74 | 80.00±0.00 | 77.52±0.43 | 71.65±0.73 | 60.42±0.90 | 55.62±5.24 | 72.67±0.20 | 67.17±0.92 |
| GLocalKD (Ma et al., 2022) | 90.59±0.61 | 86.17±0.91 | 80.59±0.00 | 73.48±0.57 | 51.42±0.66 | 51.24±0.60 | 78.94±0.00 | 70.21±0.00 |
| iGAD (Zhang et al., 2022) | 92.58±1.25 | 85.20±2.30 | 74.83±2.30 | 70.39±2.60 | 72.09±2.29 | 61.94±1.09 | 80.56±2.57 | 74.57±2.45 |
| SIGNET (Liu et al., 2023a) | 87.73±2.45 | 73.07±4.11 | 59.53±3.45 | 56.76±3.47 | 52.80±2.53 | 20.24±4.92 | 77.02±1.07 | 77.06±1.70 |
| MUSE (Kim et al., 2024) | 83.81±5.17 | 75.36±5.02 | 61.06±3.03 | 58.32±3.08 | 54.14±3.23 | 52.14±3.49 | 31.07±4.58 | 35.67±4.68 |
| DO2HSC (Zhang et al., 2024) | 88.83±6.58 | 86.80±6.21 | 77.12±2.15 | 70.87±2.73 | 63.16±3.36 | 58.36±2.95 | 68.31±4.31 | 66.63±3.04 |
| AGDiff | **95.83±2.15** | **89.45±1.37** | **88.23±0.67** | **84.06±0.59** | **77.59±3.39** | **68.15±1.49** | **91.21±1.84** | **86.04±2.26** |

*Table 2.* Average AUCs and F1-Scores with standard deviation (10 trials) on four large-scale imbalanced graph datasets. The best results are marked in **bold**, and "N/A" denotes the result is unavailable.

| Method | SW-620 | | MOLT-4 | | PC-3 | | MCF-7 | |
|---|---|---|---|---|---|---|---|---|
| | AUC | F1-Score | AUC | F1-Score | AUC | F1-Score | AUC | F1-Score |
| OCGTL (Qiu et al., 2022) | 67.69±0.02 | 27.01±0.90 | 57.42±2.38 | 53.38±0.64 | 68.42±1.73 | 27.03±0.42 | 64.92±1.92 | 34.81±1.70 |
| GLocalKD (Ma et al., 2022) | 64.14±0.92 | 60.73±0.03 | 63.43±1.26 | 60.73±0.03 | 66.08±0.04 | 43.13±0.14 | 61.43±1.26 | 45.00±0.17 |
| iGAD (Zhang et al., 2022) | 85.82±0.69 | 63.68±1.56 | 83.59±1.07 | 63.30±1.17 | 86.04±1.14 | 63.50±0.73 | 83.22±0.64 | 64.70±2.58 |
| SIGNET (Liu et al., 2023a) | 39.32±0.77 | 75.40±0.19 | 44.28±0.33 | 70.28±0.16 | 40.56±3.05 | 76.17±0.31 | 40.22±0.55 | 68.30±0.42 |
| MUSE (Kim et al., 2024) | N/A | N/A | N/A | N/A | 49.18±2.42 | 76.60±0.71 | 48.78±2.01 | 68.87±0.99 |
| DO2HSC (Zhang et al., 2024) | 43.12±0.70 | 33.65±0.66 | 51.51±2.39 | 42.30±1.34 | 52.25±3.18 | 35.66±1.26 | 53.08±2.38 | 43.73±1.32 |
| AGDiff | **91.60±0.30** | **90.97±0.13** | **87.75±0.06** | **87.22±0.01** | **94.32±2.19** | **93.12±1.83** | **89.38±0.86** | **87.25±0.39** |

standard deviations, which are computed over 10 independent runs of each algorithm to ensure reliable evaluation.

### 5.2. Comparison with State-of-the-arts Baselines

#### 5.2.1. EXPERIMENT ON MODERATE-SCALE GLAD

Table 1 presents the experimental results of AGDiff and other baseline methods on moderate-scale graph datasets, where AGDiff consistently outperforms state-of-the-art graph kernel and GNN-based GLAD methods across all datasets. For instance, on the DD dataset, AGDiff achieves a remarkable AUC of 88.23% and an F1-score of 84.06%, significantly outperforming advanced baselines such as SIGNET (AUC: 70.67%, F1-Score: 59.44%), MUSE (AUC: 61.06%, F1-Score: 58.32%), and DO2HSC (AUC: 66.04%, F1-Score: 61.14%). While graph kernel methods, such as RW, show promising results on certain datasets, they usually suffer from scalability limitations, as seen with RW running out of memory on DD. Moreover, we can observe that the semi-supervised iGAD method generally achieves strong performance compared to other baselines due to its access to partially labeled data. Nevertheless, AGDiff, despite be-

ing unsupervised, demonstrates competitive performance against iGAD. The superior performance of AGDiff can be attributed to the leverage of a latent diffusion model, which generates pseudo-anomalous graphs to enrich the training process with diverse potential anomalous patterns. As a result, AGDiff can learn fine-grained distinctions between normal and anomalous graphs without relying on reconstruction or labeled data, making it a robust and scalable solution for graph-level anomaly detection.

#### 5.2.2. EXPERIMENT ON LARGE-SCALE IMBALANCED GLAD

Table 2 presents the experimental results of AGDiff and other baseline methods on large-scale imbalanced graph datasets. A key observation from the results is the metric-specific strengths of certain baseline methods. For example, on SW-620, SIGNET achieves a high F1-Score (75.40%) but a low AUC (39.32%), while OCGTL achieves a high AUC (67.69%) but exhibits a relatively low F1-Score (27.01%). In contrast, we can observe that AGDiff consistently achieves superior performance on both metrics

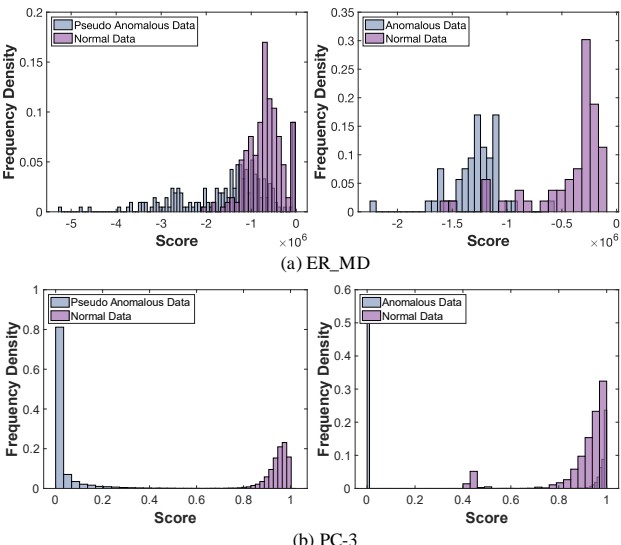

*Figure 2.* The scoring distribution of AGDiff on ER_MD and PC-3. The left and the right columns represent the results of the training stage and testing stage, respectively. Note that the $x$-axis and $y$-axis indicate the output scores and the number density of data samples within a certain interval.

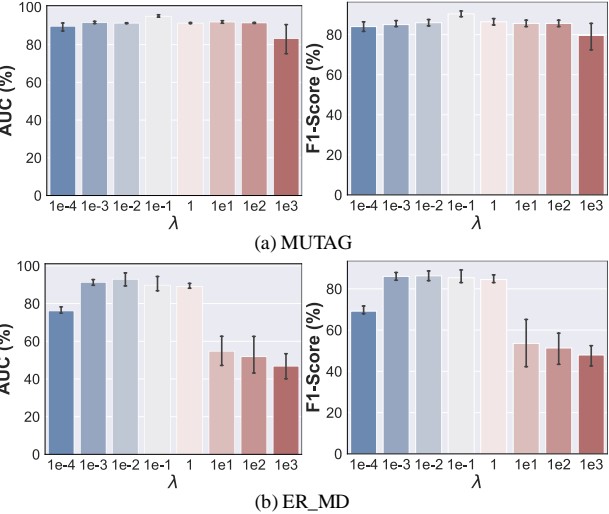

*Figure 3.* Anomaly detection performance (AUC and F1-Score) on MUTAG and ER_MD under different $\lambda$ values.

across all datasets. Additionally, we observe that the semi-supervised method, iGAD, significantly outperforms other baselines (*e.g.,* 86.04% AUC on PC-3) as it leverages partial real anomalies to refine its decision boundary. Nevertheless, AGDiff still surpasses iGAD despite being an unsupervised method across all datasets. For example, AGDiff achieves AUCs of 91.60% and 94.32% on SW-620 and PC-3, respectively, which improves by 5.78% and 8.28% over iGAD. The reason for this improvement is that although iGAD utilizes a certain amount of labeled anomalies to enhance the decision boundary learning, they may not be sufficiently representative of real anomalies due to the limited availability and diversity of anomalies in a large-scale imbalance scenario. Different from semi-supervised iGAD, the diverse pseudo-anomalous graphs generated from AGDiff provide rich self-supervised signals for training. This enables the anomaly detector to distinguish between normal and anomalous graphs with subtle deviations, which effectively overcomes the limited diversity of available anomalies in imbalanced scenarios.

### 5.3. Scoring Distribution Analysis

To verify the effectiveness of the pseudo-anomalous graphs generated by AGDiff in facilitating anomaly detection, we analyze the scoring distributions of normal, pseudo-anomalous, and real anomalous graphs on the ER_MD and PC-3 datasets, as shown in Figure 2. In ER_MD, pseudo-anomalous graphs exhibit a certain overlap with normal graphs, which indicates that generated pseudo-graph anoma-

lies maintain structural similarities to normal graphs while introducing subtle deviations via the latent diffusion process. In contrast, real anomalous graphs are more distinctly separated from normal graphs, which implies that incorporating pseudo-anomalous graphs during training improves the discriminative ability of the anomaly detector to identify real anomalies. Moreover, in PC-3, we observe that the scoring distribution of pseudo-anomalous graphs primarily lies in a low-score region, whereas real anomalous graphs remain well-separated from normal graphs in the test stage. This separation highlights the role of AGDiff in modeling the diversity of potential anomalies, which generates pseudo-graph anomalies to provide rich anomalous signals to facilitate robust decision boundary learning, particularly in large-scale imbalanced scenarios.

### 5.4. Parameter Analysis

To evaluate the impact of key hyper-parameters on the anomaly detection performance of AGDiff, we conduct a sensitivity analysis on hyper-parameter $\lambda$, which balances the contribution of the diffusion loss. Figure 3 shows the trend of AUC and F1-Score under different values of $\lambda$ on MUTAG and ER_MD. As shown in the figure, a moderate $\lambda$ generally improves performance as it ensures AGDiff generates informative pseudo-anomalous graphs, which in turn enhances the discriminative ability of the model. However, excessively large $\lambda$ prioritizes diffusion modeling over classification, which tends to make the generated pseudo-anomalous graphs overly resemble normal graphs, thereby increasing the difficulty of training the anomaly detector. Conversely, a very small $\lambda$ can also degrade the anomaly detection performance because it limits the diversity of the generated pseudo anomalies.

*Table 3.* Ablation study results on MUTAG and ER_MD.

| Method | MUTAG | | ER_MD | |
|---|---|---|---|---|
| | AUC | F1-Score | AUC | F1-Score |
| **w/o** Pre-train | 71.76±26.65 | 72.22±27.21 | 64.10±9.87 | 61.49±9.54 |
| **w/o** Condition | 63.62±38.32 | 65.19±37.44 | 68.40±0.21 | 65.19±0.60 |
| **w/o** Latent Diffusion | 48.16±4.46 | 50.67±1.89 | 63.88±3.07 | 58.48±1.54 |
| AGDiff | **95.83±2.15** | **89.45±1.37** | **91.21±1.84** | **86.04±2.26** |

## 5.5. Ablation Study

We conduct an ablation study to analyze the impact of each component in AGDiff, including the pre-training strategy, condition embedding, and the latent diffusion module. Particularly, we utilize the reconstruction error to detect anomalies when we remove the conditioned latent diffusion module, as pseudo-anomalous graphs are not available in this variant. Table 3 presents the experimental results on MUTAG and ER_MD, where we can have the following observations. First, the significant performance degradation when the pertaining strategy indicates the importance of pre-training to our framework, as it provides a well-structured latent space for modeling the normality, so that the model is able to generate informative pseudo-anomalous graphs. Second, we can observe that removing the condition embedding also significantly degrades the anomaly detection performance. This is because the condition embedding injects additional variability via introducing a noise-augmented condition vector, which encourages the model to generate diverse pseudo-anomalous samples that better capture the nuances of anomalous patterns. Otherwise, the generated graphs may closely resemble normal graphs, which leads to the difficulty of distinguishing them. Lastly, we observe the most severe performance decline in the reconstruction-based variants, which highlights the limitations of reconstruction-based methods, which generally struggle to differentiate between normal and anomalous graphs when structural deviations are subtle. In contrast, our approach leverages the latent diffusion process to explicitly model pseudo-anomalous variations, providing more robust and discriminative learning signals for training the anomaly detector.

## 5.6. More Experimental Analysis

We provide a more experimental analysis of the proposed AGDiff method in the Appendix, such as the algorithm analysis (**Appendix** A), visualization results (**Appendix** D), more parameter analysis (**Appendix** E), and more ablation study (**Appendix** F), etc.

## 6. Conclusion

In this work, we introduced Anomalous Graph Diffusion (AGDiff), a novel framework that addresses the scarcity of anomalous data in graph-level anomaly detection. By introducing a latent diffusion module to inject controlled perturbations into graph representations, AGDiff is able to generate diverse pseudo-anomalous graphs that closely resemble normal ones while exhibiting subtle deviations. A GNN-based anomaly detector is then jointly trained with the latent diffusion module to distinguish the pseudo-anomalous and normal graphs. Moreover, we theoretically demonstrated the effectiveness of these perturbed graphs generated via AGDiff for facilitating the learning of a more robust decision boundary. Empirical evaluations on multiple graph benchmarks validate the superiority of AGDiff against state-of-the-art GLAD baselines. Two limitations of this work are: (1) It assumes a sufficiently representative distribution of normal graphs, which may not hold in shifting or highly heterogeneous environments. (2) While AGDiff can generate pseudo graph anomalies to enhance decision boundary training, it is currently limited to static graphs. Future research can focus on exploring more flexible noise scheduling diffusion approaches, as well as investigating the challenging GLAD tasks in heterogeneous environments or dynamic graph settings.

## Acknowledgements

This research is supported by the National Research Foundation Singapore and DSO National Laboratories under the AI Singapore Programme (AISG Award No: AISG2-RP-2020-018).

## Impact Statement

This paper presents work whose goal is to advance the field of Machine Learning. There are many potential societal consequences of our work, none of which we feel must be specifically highlighted here.

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

# A. Algorithm Analysis

**Theoretical Analysis.** Here we introduce two algorithms to achieve anomaly detection when given normal data. Specifically, when given a normal dataset $\mathcal{G}^{\text{normal}}$, the algorithm $\mathcal{A}_1$ first generates a number of normal data $\text{Gen}(\mathcal{G}^{\text{normal}})$ with some generation method $\text{Gen}(\cdot)$ (*e.g.*, diffusion process) by the given normal data $\mathcal{G}^{\text{normal}}$, then whether a new dataset $\mathcal{G}$ is anomalous is determined by the data divergence $d(\mathcal{G}, \mathcal{G}^{\text{normal}})$. If $d(\mathcal{G}, \mathcal{G}^{\text{normal}}) > d(\text{Gen}(\mathcal{G}^{\text{normal}}), \mathcal{G}^{\text{normal}})$, then $\mathcal{G}$ is classified as anomalous data. For the algorithm $\mathcal{A}_2$, it first perturbs the normal dataset $\mathcal{G}^{\text{normal}}$ to get a perturbed dataset $\widetilde{\mathcal{G}}^{\text{normal}}$, then use the same generation method $\text{Gen}(\cdot)$ to generate a number of perturbed data $\text{Gen}(\widetilde{\mathcal{G}}^{\text{normal}})$. This generated perturbed data is regarded as auxiliary anomalous data, and finally $\mathcal{A}_2$ uses a classifier to obtain a decision boundary for $\mathcal{G}^{\text{normal}}$ and $\text{Gen}(\widetilde{\mathcal{G}}^{\text{normal}})$. The criterion for $\mathcal{A}_2$ becomes that if $d(\mathcal{G}, \mathcal{G}^{\text{normal}}) > d(\mathcal{G}, \text{Gen}(\widetilde{\mathcal{G}}^{\text{normal}}))$, then $\mathcal{G}$ is classified as anomalous data.

Notice that it is reasonable to assume under the same generation method $\text{Gen}(\cdot)$, the reconstruction error between the original data and the generated data does not change, *i.e.*, $d(\text{Gen}(\mathcal{G}^{\text{normal}}), \mathcal{G}^{\text{normal}}) = d(\widetilde{\mathcal{G}}^{\text{normal}}, \text{Gen}(\widetilde{\mathcal{G}}^{\text{normal}}))$. Therefore, if

$$d(\widetilde{\mathcal{G}}^{\text{normal}}, \text{Gen}(\widetilde{\mathcal{G}}^{\text{normal}})) > d(\mathcal{G}, \text{Gen}(\widetilde{\mathcal{G}}^{\text{normal}})),$$

then once $\mathcal{A}_1$ successfully detects the anomalous data, $\mathcal{A}_2$ can also successfully detect the anomalous data, which implies that $\mathcal{A}_2$ is better than $\mathcal{A}_1$ in detecting the anomalous data.

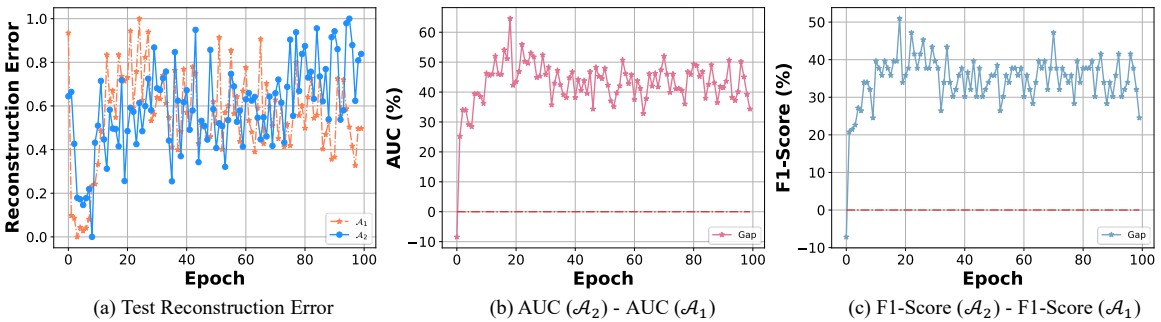

(a) Test Reconstruction Error  (b) AUC $(\mathcal{A}_2)$ - AUC $(\mathcal{A}_1)$  (c) F1-Score $(\mathcal{A}_2)$ - F1-Score $(\mathcal{A}_1)$

*Figure 4.* Empirical study results, including the reconstruction error trends and performance gap (both AUC and F1-Score) between $\mathcal{A}_1$ and $\mathcal{A}_2$ on ER_MD dataset.

**Empirical Analysis.** We empirically validate our theoretical analysis through comprehensive experiments on the ER_MD dataset, examining both reconstruction error dynamics and anomaly detection performance over 100 training epochs. Our evaluation metrics include reconstruction error, AUC, and F1-Score on the test set. Figure 4 presents the results of the **test set** across training epochs for $\mathcal{A}_1$ and $\mathcal{A}_2$, where we have the following key observations:

- The reconstruction error of $\mathcal{A}_1$ decreases in the early training stages while gradually increasing afterward. This trend suggests overfitting to normal data, where $\mathcal{A}_1$ effectively learns to reconstruct normal graphs, leading to a sharp reduction in reconstruction error. This pattern indicates that while $\mathcal{A}_1$ effectively learns to reconstruct normal graphs in early stages, it subsequently overfits the training distribution, compromising its generalization capability on unseen test data, which can be observed in the performance gap changes in AUC and F1-Score.

- $\mathcal{A}_2$ consistently exhibits a higher and fluctuating reconstruction error throughout training. This observation suggests that the introduction of pseudo-anomalous graphs prevents overfitting to normal patterns, which makes the model learn a more discriminative representation. Unlike $\mathcal{A}_1$ which primarily minimizes reconstruction loss, $\mathcal{A}_2$ actively differentiates between normal and anomalous graphs, as reflected in the AUC gap: while initially lagging behind $\mathcal{A}_1$, $\mathcal{A}_2$ steadily improves and eventually surpasses $\mathcal{A}_1$ as its classifier learns to leverage pseudo-anomalous data for decision boundary refinement.

- Since anomalous graphs inherently exhibit higher reconstruction errors, an effective anomaly detection model should maintain a sufficient error gap between normal and abnormal samples. This is validated by the AUC trends, where $\mathcal{A}_2$ eventually surpasses $\mathcal{A}_1$, demonstrating that its classifier progressively refines its decision boundary by effectively

incorporating pseudo-anomalous graphs. The stabilization of the AUC gap in later epochs suggests that $\mathcal{A}_2$ converges to a robust and generalizable anomaly detection model, whereas $\mathcal{A}_1$ remains constrained by its reliance on reconstruction.

These empirical results validate our theoretical analysis that as the reconstruction error increases, our method becomes more robust. Specifically, when the reconstruction error is small, both $\mathcal{A}_1$ and $\mathcal{A}_2$ achieve similar anomaly detection performance. While as the reconstruction error increases, $\mathcal{A}_2$ significantly outperforms $\mathcal{A}_1$, which matches our theoretical results that decision boundary learning via pseudo-anomalous graphs leads to superior anomaly detection. Compared to the purely reconstruction-based strategy of $\mathcal{A}_1$, the explicit optimization of decision boundaries through pseudo-anomalous graphs in $\mathcal{A}_2$ offers a more robust solution for graph-level anomaly detection.

## B. Datasets

Table 4 presents the details of the graph datasets used in our experiments, including the number of graphs, average nodes, average edges, node classes, and graph classes. Additionally, we also provide the anomaly ratio and data type of each dataset in the table.

*Table 4.* Detailed information of the graph benchmark datasets.

| Dataset Name | # Graphs | # Average $[V]$ | # Average $[E]$ | # Node Classes | # Graph Classes | Anomalous Rate | Data Types |
|---|---|---|---|---|---|---|---|
| MUTAG | 188 | 17.93 | 19.79 | 7 | 2 | 33.51% | Molecule |
| COX2 | 467 | 41.22 | 43.45 | 8 | 2 | 21.84% | Molecule |
| ER_MD | 446 | 21.33 | 234.85 | 10 | 2 | 40.58% | Molecule |
| DD | 1,178 | 284.32 | 715.66 | 82 | 2 | 41.34% | Biology |
| SW-620 | 40,532 | 26.06 | 28.09 | 65 | 2 | 5.95% | Molecule |
| MOLT-4 | 39,765 | 26.10 | 28.14 | 64 | 2 | 7.90% | Molecule |
| PC-3 | 27,509 | 26.36 | 28.49 | 45 | 2 | 9.34% | Molecule |
| MCF-7 | 27,770 | 26.40 | 28.53 | 46 | 2 | 8.26% | Molecule |

## C. Implementation Details

Here we provide the implementation details in our experiment, including the data split, network architecture, hyper-parameter setting, baseline setting, and computing resources as follows:

- **Data Split:** For small to moderate-scale datasets, we follow the data split method used in (Zhang et al., 2024). Specifically, 80% of the normal data is used for training, while the test set combines the remaining normal samples with an equal or smaller number of anomalies. For large-scale imbalanced datasets, 80% of the normal data is allocated for training, and the test set includes the rest of the normal data along with all anomalies.

- **Network Architecture:** Following prior works (Ma et al., 2022; Zhang et al., 2024), we adopt the Graph Isomorphism Network (GIN) (Xu et al., 2019) as the backbone. Both the pre-trained graph representation learning model and the anomaly detector are constructed with 3 GIN layers. Additionally, we employ a 2-layer MLP for both the perturbation condition model and the projector within the anomaly detector. The latent dimension is set to 16 for moderate-scale datasets and 512 for large-scale datasets.

- **Hyper-parameter Setting:** We adopt Adam (Kingma & Ba, 2014) optimizer with fixed learning rate $\rho = 0.001$ for both pre-training and training stages. We first pre-train the graph generator for 100 epochs, with the KL-divergence loss coefficient set to 0.001. The diffusion process employs a fixed time step of $T = 1,000$ for all datasets. In training, we adopt a batch size of 16 for moderate-scale datasets and 512 for large-scale datasets. The full model is trained for 200 epochs. To balance the contribution of the diffusion loss, we tune the coefficient $\lambda$ via a grid search over the range $[0.0001, 1000]$ to achieve optimal performance. We also evaluate the impact of $\lambda$ in Section 5.4.

- **Baseline Setting:** We access the official codes of all graph kernel methods from the GraKel repository (Siglidis et al., 2020), where we extract the kernel matrix on each dataset and apply OCSVM (via Scikit-learn (Pedregosa et al., 2011)) to achieve anomaly detection. For the GNN-based GLAD baselines, we reproduce all the results of the compared method by reproducing their officially released codes. Particularly, we employ the same architecture of the backbone network and the data split strategy as the proposed AGDiff to ensure a fair comparison.

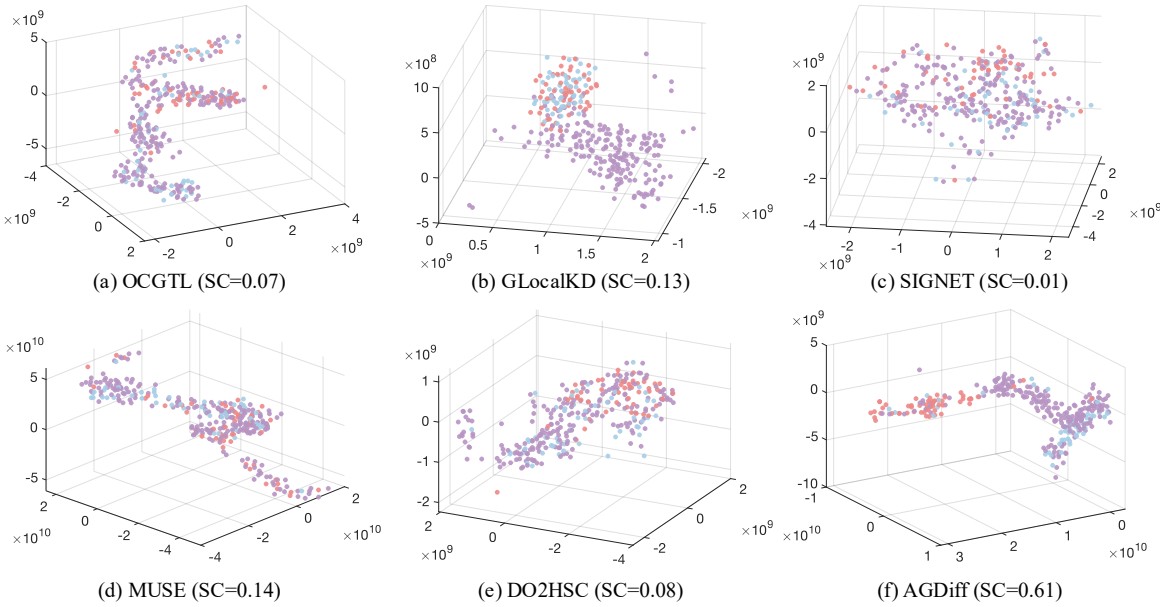

*Figure 5.* The t-SNE visualization comparison on ER_MD dataset, where the data points marked in purple, blue, and red represent training normal graphs, testing normal graphs and anomalous graphs, respectively. SC denotes the Silhouette Coefficient metric, which is used for evaluating.

- **Computing Resources:** All the experiments in this paper are performed on the NVIDIA Tesla H100 GPU (80GB) with Intel Xeon Platinum 8480CL CPU.

## D. Visualization Results

To analyze how different methods capture graph representations, we employ t-SNE (Van der Maaten & Hinton, 2008) to visualize the latent embeddings of AGDiff and several baseline methods for comparison. As shown in Figure 5, we can observe that the latent embeddings of AGDiff exhibit a more compact and well-separated structure, with normal and anomalous graphs forming distinct regions in the latent space. This observation suggests that our model effectively simulates subtle deviations from normality. Moreover, to quantitatively evaluate the effectiveness of the learned representations in AGDiff, we further calculate the Silhouette Score (SC) between normal and anomalous embeddings for each method. Note that the value of SC ranges from $[-1, 1]$, and a higher SC indicates a better separation. From this table, we can observe that AGDiff achieves an SC of 0.61, which is significantly higher than state-of-the-art GLAD baselines such as MUSE (SC = 0.14), DO2HSC (SC = 0.08), and SIGNET (SC = 0.01). This observation indicates a better separation between normal and anomalous embeddings from a statistical perspective. These results highlight the effectiveness of our approach in learning expressive and anomaly-aware graph representations for anomaly detection.

In order to more intuitively understanding the generated pseudo graphs, we randomly pick four pairs of generated pseudo graphs and normal graphs to visualize their graph structures, as shown in Figure 6. These pseudo graphs exhibit noticeable structural deviations from their normal counterparts, such as the varied density and sparser connections in different regions, and modified local connectivity patterns around nodes. For example, some generated graphs exhibit sparser regions or more peripheral nodes (*e.g.*, the first and fourth pseudo graphs in the second row). These visual discrepancies are critical as they empirically validate the rationale of these generated pseudo graph anomalies in helping train the anomaly detector. In particular, we further utilize the normalized Laplacian spectral distance to quantify the discrepancy between the generated anomalies and normal graphs. Specifically, we computed pair-wise spectral discrepancies between: (1) Generated pseudo-anomalous graphs and normal graphs, which ranged in $[0.17, 1.12]$, and (2) Real anomalous graphs and normal graphs, which ranged in $[0.40, 5.12]$. And the corresponding Laplacian spectral distances of the examples in Figure 6 are larger than the minimum pair-wise discrepancy of real anomalies. While the range for pseudo anomalies is narrower compared to real anomalies, this outcome aligns with our design intent. Rather than mimicking the extreme deviations observed in real anomalies, we aimed to produce controlled yet challenging perturbations to enhance the decision boundary learning. These

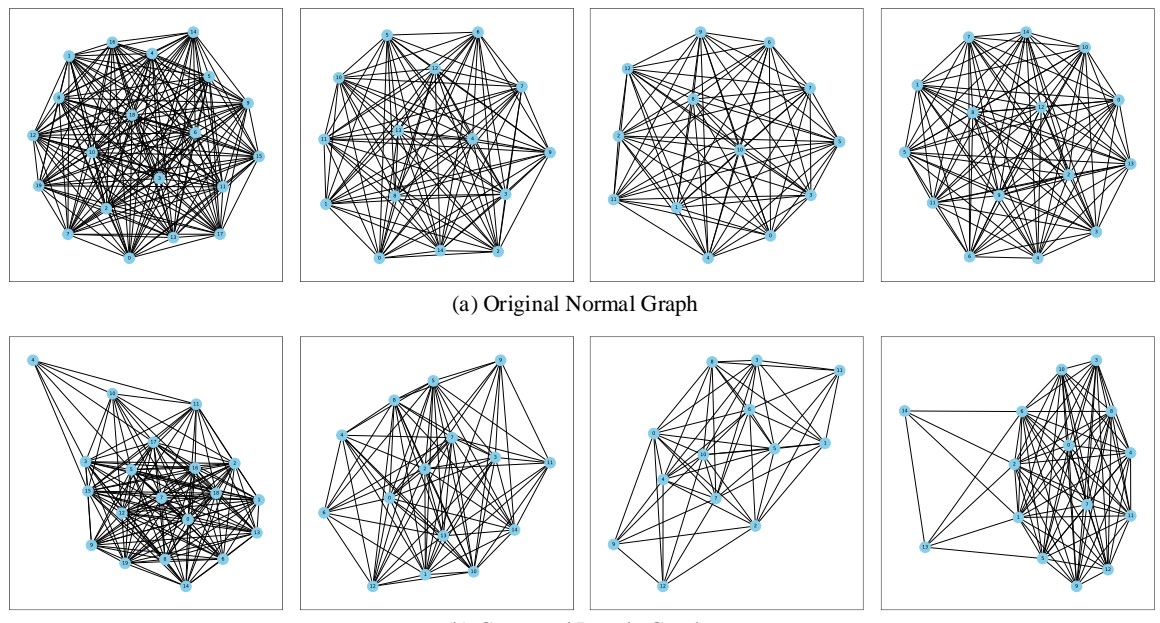

(a) Original Normal Graph

(b) Generated Pseudo Graph

*Figure 6.* The visual comparison of normal and generated pseudo graphs pairs on the ER_MD dataset. Note that the normalized Laplacian spectral distances between each pair in this figure are 0.44, 0.43, 0.62, and 0.43, respectively.

results indicate that the generated pseudo anomalies exhibit a meaningful discrepancy from the normal graphs, providing sufficient diversity to effectively challenge the model during training.

## E. More Parameter Analysis

**Effect of Different Noise Magnitudes $\eta$.** To evaluate the effect of the noise magnitude parameter $\eta$ on the performance of AGDiff, we conduct an experiment by varying it from a wide range $[0, 100]$. Table 5 presents the experimental results. We can observe a performance degradation when this noise term is removed (*i.e.*, $\eta = 0$) or set to an excessive value (*e.g.*, $\eta = 10$). Our explanation for these observations is that (1) removing the noise term may lead to over-proximity between the pseudo graph anomalies and the normal graph, thus making it difficult to train an anomaly detector. (2) While an excessively high $\eta$ can lead to pseudo anomalies that deviate too far from normal data, thus reducing the discriminative ability of the anomaly detector. These findings highlight the importance of balancing $\eta$ to ensure meaningful perturbation to the initial latent representation.

*Table 5.* Anomaly detection performance of different noise magnitudes $\eta$ on MUTAG and ER_MD. The best results are marked in **bold**.

| Datasets | Metrics | $\eta = 0$ | $\eta = 0.01$ | $\eta = 0.1$ | $\eta = 1$ | $\eta = 10$ | $\eta = 100$ |
|---|---|---|---|---|---|---|---|
| MUTAG | AUC | 93.20±2.64 | 92.64±0.32 | 92.40±0.08 | **95.83±2.15** | 92.00±0.00 | 92.00±0.00 |
| | F1-Score | 86.00±2.00 | 88.00±0.00 | 88.00±0.00 | **89.45±1.37** | 88.00±0.00 | 88.00±0.00 |
| ER_MD | AUC | 82.95±2.17 | 86.81±0.91 | 87.78±1.07 | **91.21±1.84** | 82.08±0.52 | 81.76±0.59 |
| | F1-Score | 77.36±1.89 | 80.19±2.83 | 85.09±0.94 | **86.04±2.26** | 71.70±0.00 | 72.64±0.94 |

**Effect of Different Time Steps $T$ in Diffusion Sampling.** Here we also analyze the performance fluctuations when setting different time steps $T$ on the MUTAG and ER_MD datasets. It can be observed that the performance (e.g., on MUTAG) generally improves from $T = 250$ (91.52% AUC) to $T = 750$ (95.84% AUC) or $T = 1000$ (95.83% AUC), with performance plateauing or slightly declining when $T \geq 1000$. A trend was similarly observed on ER_MD, which indicates that a low $T$ may be insufficient to generate high-quality pseudo graphs. Nevertheless, a high $T$ can also lead to

over-diffusion, making the generated graphs excessively similar to normal graphs, which in turn diminishes the capability of the anomaly detector to differentiate them and increases computational cost as well. These findings highlight the importance of appropriately selecting $T$ to balance the generation quality and computational cost.

*Table 6.* Anomaly detection performance (AUCs and F1-Scores) of different time steps $T$ on MUTAG and ER_MD in diffusion sampling process. The best results are marked in **bold**.

| Datasets | Metrics | $T = 250$ | $T = 500$ | $T = 750$ | $T = 1000$ | $T = 1250$ | $T = 1500$ |
|---|---|---|---|---|---|---|---|
| MUTAG | AUC | 91.52±7.84 | 95.20±3.20 | **95.84±3.84** | 95.83±2.15 | 92.00±0.00 | 92.00±0.00 |
| | F1-Score | 88.00±4.00 | 92.00±0.00 | **94.00±2.00** | 89.45±1.37 | 87.00±0.00 | 87.00±0.00 |
| ER_MD | AUC | 86.61±4.38 | 89.30±0.33 | 84.48±2.17 | **91.21±1.84** | 82.34±3.13 | 82.18±0.91 |
| | F1-Score | 78.30±6.60 | 83.02±1.89 | 79.25±5.66 | **86.04±2.26** | 73.58±1.89 | 73.58±1.89 |

## F. Ablation Study on the Number of Generated Graphs

In our initial experimental design, we set the number of generated pseudo-anomalous graphs equal to the number of normal graphs. This choice was based on the rationale that a balanced data configuration would likely foster more stable training dynamics and prevent the model from being biased towards the normal class. Here, we further conduct an ablation study on the anomaly ratio $r$, which represents the proportion of generated pseudo graphs relative to the quantity of normal graphs available in the training set. Table 7 shows the experimental results on the DD dataset, which offers a key insight into our anomaly detection framework under an imbalanced setting (Fang et al., 2025). The results show that a balanced data composition (*i.e.*, $r = 100\%$) is crucial for robust learning and yields the best performance. In addition, the observed trend, where performance (AUC) and stability (represented by the standard deviation) are improved from $85.06 \pm 3.42$ ($r = 30\%$) to $88.23 \pm 0.67$ ($r = 100\%$). This implies that the effective learning of the anomaly boundary heavily relies on the richness and sufficiency of the pseudo-anomalous graphs. Insufficiently generated graphs (*e.g.*, $r < 60\%$) appear to create an "information bottleneck", which limits the ability of the model to generalize the concept of anomaly from an undersampled and less diverse space.

*Table 7.* Anomaly detection performance under different anomaly ratios $r$. The best results are marked in **bold**.

| Dataset | Metrics | $r = 30\%$ | $r = 60\%$ | $r = 90\%$ | $r = 100\%$ |
|---|---|---|---|---|---|
| DD | AUC | 85.06±3.42 | 86.72±0.70 | 87.30±0.49 | **88.23±0.67** |
| | F1-Score | 80.78±4.48 | 83.23±1.45 | 84.03±0.36 | **84.06±0.59** |

