# OpenReview forum: "Leveraging Diffusion Model as Pseudo-Anomalous Graph Generator for Graph-Level Anomaly Detection"
_ICML.cc/2025/Conference — ICML 2025 spotlightposter_

### Official Review · Reviewer_2Zh2 · 2025-03-08

**Overall Recommendation:** 3

**Summary:**

In this paper, the authors proposed a novel GLAD method named AGDiff. AGDiff leverages diffusion models to generate pseudo-anomalous graphs, which addresses the challenge of anomaly scarcity in GLAD. Particularly, a latent diffusion process with perturbation conditions is proposed to generate diverse pseudo graph anomalies, and a joint training scheme is employed to integrate the training of pseudo anomaly generation and anomaly detection. Experiments on moderate-scale and large-scale imbalanced GLAD tasks demonstrated the effectiveness of AGDiff.

**Claims And Evidence:**

Yes. The key claims of this paper are well-supported by convincing evidence. For example, (1) Comparison results (Tables 1 and 2) demonstrate the consistent superiority of AGDiff over existing baselines across diverse datasets. (2) Scoring distribution analysis (Figure 2) shows overlap between normal and pseudo-anomalous graphs, with real anomalies distinctly separated, validating the claim that the generated pseudo-anomalous graphs resemble normal graphs with subtle deviations.

**Essential References Not Discussed:**

This paper has adequately discussed recent works in GLAD and diffusion models, and the authors also described the connection between the proposed method with other related works, which is sufficient to understand the key contribution of the paper.

**Experimental Designs Or Analyses:**

Yes. The experimental design is technically sound, with fair comparisons to graph kernels and GNN-based GLAD baselines. The authors conducted experiment on different GLAD scenarios and compared the proposed AGDiff with latest GLAD methods, which make the experimental results convincing.

**Methods And Evaluation Criteria:**

Yes. The proposed method is well-motivated, the latent diffusion process and joint training paradigm address the challenge of anomaly scarcity in GLAD problem. The evaluation criteria used in the experiment are suitable for GLAD.

**Other Comments Or Suggestions:**

The limitations of this work are not discussed in the conclusion. The authors are encouraged to provide additional discussion.

**Other Strengths And Weaknesses:**

*Pros:*
- The paper is well-written, and the motivation is clearly illustrated. This work bridges a gap between diffusion modeling and graph-level anomaly detection, offering a novel perspective in the field.
- The proposed perturbation conditioned latent diffusion process enables the model to eliminate reliance on labeled anomalies, making it more applicable to real-world domains.
- Extensive experiments were conducted to demonstrate the effectiveness of the proposed method. Particularly, AGDiff exhibited superior performance in the large-scale imbalanced GLAD compared to the latest baselines.

*Cons:*
- In Eq. (10), the condition vector $\textbf{c}$ is generated by perturbing the input. However, it is unclear what role $\textbf{c}$ plays in the generation of the pseudo-anomalous graphs.
- The difference between the proposed AGDiff and the diffusion-based anomaly detection approaches should be elaborated.
- The experiment fixes the use of a diffusion process with $T=1,000$ steps, while the influence of varying $T$ on the anomaly detection performance is unclear.
- The reproducibility of this work is relatively limited as the authors did not provide the implementation code for AGDiff.

**Questions For Authors:**

See Other Strengths And Weaknesses.

**Relation To Broader Scientific Literature:**

AGDiff bridges diffusion models and GLAD by leveraging a conditioned latent diffusion process to solve the anomaly scarcity problem in GLAD. It also contrasts with reconstruction-based GLAD methods, such as MUSE, by explicitly generating pseudo-anomalies.

**Theoretical Claims:**

Yes. The theoretical analysis in Appendix A intuitively demonstrated the superiority of the proposed method via the comparison between two algorithms (\text{A}_{1} and \text{A}_{2}).

---

> ### Author Rebuttal · Authors · 2025-03-31
>
> **We thank the reviewer for recognizing our work. Below are our responses to your concerns.**
>
> **To W1:** The condition vector $\mathbf{c}$ is crucial for ensuring the generation quality of pseudo graph anomalies. By perturbing the initial latent embedding through a learnable perturbation transformation, $\mathbf{c}$ injects additional variability into the latent diffusion process. This ensures that the denoising network is conditioned to deviate from purely "normal" reconstructions. We have evaluated the influence of $\mathbf{c}$ via an Ablation Study (Table 3), where removing $\mathbf{c}$ led to a substantial performance drop. In that case, the generated graphs may closely resemble normal graphs, making it more difficult for the anomaly detector to distinguish them and thus reducing its overall discriminative capability.
>
> **To W2:** AGDiff differs fundamentally from existing diffusion model-based methods [1,2,3]. Whereas existing approaches focus on modeling data normality, treating anomalies as outliers with significant reconstruction errors, AGDiff leverages the diffusion model to simulate subtle deviations from normal data and adaptively generate diverse pseudo-anomalous graphs that provide rich anomalous signals for refining the decision boundary learning. This proactive generation strategy facilitates the training of a powerful anomaly detector (as evidenced by the performance comparisons in Tables 1\&2), enabling the model to better distinguish between normal and real anomalous graphs. We have included some discussions in Section 2.2, and we will elaborate further in the paper to emphasize our contributions.
> ```
> [1] Wolleb, J., et al. Diffusion models for medical anomaly detection, MICCAI, 2022.
> [2] Tebbe, J., et al. Dynamic addition of noise in a diffusion model for anomaly detection, CVPR, 2024.
> [3] Li, J., et al. Diffgad: A diffusion-based unsupervised graph anomaly detector, ICLR 2025.
> ```
>
> **To W3:** To clarify the effect of $T$ on the model's performance, we have conducted additional experiments by varying $T$ from 250 to 1500. The experimental results are as follows:
>
> $\begin{matrix}
> \\hline
> \text{Datasets}&\text{Metrics}&T=250&T=500&T=750&T=1000&T=1250&T=1500\\\\\hline
> \text{MUTAG}&\text{AUC}&91.52(7.84)&95.20(3.20)&\bf95.84(3.84)& 95.83(2.15)&92.00(0.00)&92.00(0.00)\\\\
> &\text{F1-Score}&88.00(4.00)&92.00(0.00)&\bf94.00(2.00)& 89.45(1.37)&87.00(0.00)&87.00(0.00)\\\\\hline
> \text{ER\\_MD}&\text{AUC}&86.61(4.38)&89.30(0.33)&84.48(2.17)&\bf91.21 (1.84)&82.34(3.13)&82.18(0.91)\\\\
> &\text{F1-Score}&78.30(6.60)&83.02(1.89)&79.25(5.66)&\bf86.04 (2.26)&73.58(1.89)&73.58(1.89)\\\\\hline
> \end{matrix}$
>
> As seen above, the performance (e.g., on MUTAG) generally improves from $T=250$ (91.52\% AUC) to $T=750$ (95.84\% AUC) or $T=1000$ (95.83\% AUC), with performance plateauing or declining when $T\geq 1000$. A trend was similarly observed on ER_MD, which indicates that a low $T$ may be insufficient to generate high-quality pseudo graphs. Nevertheless, a high $T$ can also lead to over-diffusion, making the generated graphs excessively similar to normal graphs, which in turn diminishes the capability of the anomaly detector to differentiate them and increases computational cost as well. These findings highlight the importance of selecting appropriate $T$ to balance the generation quality and computational cost. We will add this experiment and relevant discussions to our paper.
>
> **To W4:** We have provided the implementation code of AGDiff at https://anonymous.4open.science/r/AGDiff-137F/. This link is fully anonymous, with some pre-trained models attached for reproducibility.
>
> **To Other Comments:** We will follow your suggestion to further discuss the limitations of our work in the conclusion, such as (1) Our method assumes a sufficiently representative distribution of normal graphs, which may not hold in shifting or highly heterogeneous environments. (2) While AGDiff can generate pseudo graph anomalies to enhance decision boundary training, it is currently limited to static graphs. Future work could explore solving challenging GLAD tasks in heterogeneous environments or dynamic graph settings.

---

### Official Review · Reviewer_BUES · 2025-03-11

**Overall Recommendation:** 4

**Summary:**

This paper presents a diffusion-based method for generating pseudo-anomalous graphs to address anomaly scarcity in GLAD. It employs a latent diffusion process with perturbation conditions and a joint training scheme for anomaly detection. Experiments on both balanced and imbalanced datasets demonstrate its effectiveness.

**Claims And Evidence:**

Yes, the manuscript claims that the proposed method, which integrates an innovatively designed latent diffusion model as a perturbator to generate pseudo-anomalous graphs, outperforms reconstruction error-based approaches by leveraging these generated graphs to refine the decision boundary of the anomaly detector. Experimental results, along with the theoretical analysis in Appendix A, support this claim.

**Essential References Not Discussed:**

This paper presents a comprehensive discussion of both GLAD approaches and diffusion-based methods.

**Experimental Designs Or Analyses:**

Yes, the experimental settings are fair, and the evaluation metrics align with existing graph-level anomaly detection literature. The study includes multiple perspectives, including both moderate-scale balanced and large-scale imbalanced experiments.

**Methods And Evaluation Criteria:**

Yes, the motivation and proposed solution are well-justified. The evaluation criteria are fair and aligned with established anomaly detection benchmarks.

**Other Comments Or Suggestions:**

There are some grammar and format errors, e.g., in line 142, "Let … denotes" should be "Let … denote". Authors should double-check the grammar carefully.

**Other Strengths And Weaknesses:**

Strengths:
1) The paper is well-written, and the idea is novel (refer to the comment in **"Relation To Broader Scientific Literature"**).
2) Each module in the proposed method is well-designed with clear and justified motivation. The designed diffusion anomaly detection model is very different from the existing diffusion-based anomaly detection methods.
3) Comprehensive experimental results demonstrate the method's effectiveness.
4) Theoretical analysis is provided to support the claims.

Weaknesses:
1) How does the proposed method ensure a gap between real normal data and pseudo abnormal data? Are they definitively different? The authors should clarify this point.
2) An analysis of the computational complexity is absent.
3) In the theoretical analysis, why must A_2 perturb normal data before generating pseudo anomalous data? Why not directly use perturbed normal data to train the classifier? A more intuitive explanation, possibly with examples, would be helpful.
4) The code used for the experiments is not provided.

**Questions For Authors:**

Please refer to Weaknesses.

**Relation To Broader Scientific Literature:**

AGDiff offers a novel approach to anomaly detection. It generates pseudo-anomalous data and trains a strong classifier to precisely define the decision boundary, tightly enclosing normal data. Additionally, it leverages a diffusion model with perturbation conditions to ensure pseudo-anomalies differ from normal data. This makes the method both innovative and effective.

**Theoretical Claims:**

Yes, the theoretical analysis in Appendix A demonstrates that the proposed solution outperforms reconstruction-based models.

---

> ### Author Rebuttal · Authors · 2025-03-31
>
> **Thank you for your recognition, and we hope the responses below can solve your concerns:**
>
> **To W1:** AGDiff ensures the gap between normal and pseudo-anomalous graphs via its **controlled latent diffusion process** and **joint training**. Rather than using arbitrary noise, the conditioned vector (Eq. 10) introduces learnable perturbations that simulate potential and diverse anomalous patterns while preserving structural plausibility. There are several pieces of evidence to support this claim, e.g., (1) Varying $\eta$ influences the generation quality of pseudo graph anomalies, thereby influencing the final performance (refer to Reviewer kMMx's W2). (2) Removing the conditioning vector $\mathbf{c}$ leads to a substantial performance drop (refer to  Reviewer 2Zh2's W1). (3) Our statistical analysis (refer to  Reviewer kMMx's W4) confirms that there is indeed a meaningful gap between pseudo-anomalous graphs and normal graphs.
>
> **To W2:** Here, we provide a computational complexity analysis for AGDiff. For a dataset of $N$ graphs, each with an average of $m$ nodes (feature dimension $d$), $|E|$ edges, and latent dimension $d_{z}$, the AGDiff framework operates in three phases:
>
> 1. **Pre-training:** An $L$-layer GIN is employed as the backbone network in the pre-training, where the overall complexity is $\mathcal{O}(N L (|E|d + m d^{2}))$ due to the message aggregation ($\mathcal{O}(|E|d)$) and feature transformation ($\mathcal{O}(m d^{2})$).
> 2. **Pseudo Anomaly Generation:** This phase involves the computation of conditional vector ($\mathcal{O}(N d_{z}^{2})$) and a $T$-step latent diffusion process ($\mathcal{O}(N T m d_{z}^{2})$) across all graphs.
> 3. **Decoding \& Anomaly Detection:** Decoding involves the computation of node attributes and adjacency matrices from latent embeddings, which results in time complexity of $\mathcal{O}(N|E|d_{z})$ when we apply a negative sampling strategy in practice. The computational complexity of the anomaly detector is similar to the pre-training stage, i.e., ($\mathcal{O}(N L(|E|d + m d^{2}))$), due to their similar network structure.
>
> Therefore, the overall computational complexity of AGDiff is approximately $\mathcal{O}(N L (|E|d + m d^{2}) + N (T m+1) d_{z}^{2} + N|E|d_{z})$, which is comparable with other state-of-the-art baselines such as SIGNET, MUSE, DO2HSC. Besides, potential optimizations like using parallel computation further enhance efficiency. We will add this analysis to the paper.
>
> **To W3:** We would like to address your concern from two perspectives:
> 1. **Subtle Deviations Facilitate Decision Boundary Learning.**  In $\mathcal{A}_{2}$, perturbing normal data before the generation process aims to ensure subtle deviations of the generated pseudo-anomalous graphs from normal graphs, which is critical for effective decision boundary learning. Relevant evidence can be found in the response to Reviewer kMMx’s W2.
>
> 2. **Learnable Perturbations Expand Anomaly Diversity.** Directly using perturbed normal data as pseudo anomalies would constrain the diversity of anomalous patterns the model encounters, as the perturbations would be static and lack adaptability. This limitation may cause the overfitting of the classifier to a narrow set of pseudo anomalies, reducing its generalizability. In $\mathcal{A}_2$, however, the perturbations are learnable and dynamically adjusted during joint training with the anomaly detector. As the detector improves, the perturbation mechanism evolves, generating increasingly sophisticated and varied pseudo anomalies. This adaptive process exposes the model to a broader spectrum of potential anomalies, enhancing its robustness and generalization.
>
> **To W4:** We have provided the implementation code of AGDiff at https://anonymous.4open.science/r/AGDiff-137F/. The link is fully anonymous, and some pre-trained models are also attached for reproducibility.
>
> **To Other Comments:** We will double-check our paper and correct all identified grammatical and formatting issues such as replacing "Let … denotes" with "Let … denote".

---

> > ### Comment · Reviewer_BUES · 2025-04-04
> >
> > The authors' clarifications have addressed my concerns, and the added complexity analysis and code further strengthen the paper. I will raise my score accordingly.

---

> > > ### Author Response · Authors · 2025-04-04
> > >
> > > **Dear Reviewer BUES:**
> > >
> > > Thank you for the positive feedback. We are glad that our responses have addressed your concerns. We sincerely appreciate your valuable comments, which have helped us improve the paper. The related content will be included in the revised version.
> > >
> > > Best regards,
> > >
> > > Authors

---

### Official Review · Reviewer_kMMx · 2025-03-11

**Overall Recommendation:** 4

**Summary:**

This paper introduces Anomalous Graph Diffusion (AGDiff), a novel graph-level anomaly detection (GLAD) framework that consists of three core components: (1) a pre-training module employing variational inference to learn a structured latent space, (2) a latent diffusion process that introduces controlled perturbations to generate pseudo-anomalous graphs, and (3) a jointly trained graph anomaly detector that distinguishes normal graphs from pseudo-anomalous graphs. The central idea is to mitigate the scarcity of anomalous graphs in GLAD by generating diverse pseudo graph anomalies through a perturbation conditioned latent diffusion process. Experimental results demonstrate that AGDiff significantly outperforms state-of-the-art GLAD baselines on diverse GLAD tasks.

**Claims And Evidence:**

The advantages of AGDiff over other GLAD baselines have been validated via extensive experiments. However, the claim that the generated pseudo-anomalies of AGDiff "closely resemble normal graphs" is only qualitatively discussed via t-SNE visualizations (Appendix D). A quantitative analysis with similarity metrics, e.g., graph edit distance between generated and normal graphs would strengthen this claim.

**Essential References Not Discussed:**

The literature review is thorough, with key related works appropriately cited.

**Experimental Designs Or Analyses:**

I have checked the soundness of experimental designs and analyses of this paper. The experimental design (Tables 1 and 2) is sound and reliable, which employs consistent data splits and fair baseline comparisons.

**Methods And Evaluation Criteria:**

The proposed method makes sense for the GLAD problem. The evaluation metrics (AUC and F1-Score) are standard for GLAD, and the benchmark datasets cover both moderate-scale and large-scale imbalanced scenarios, which make sense for real-world GLAD problem.

**Other Comments Or Suggestions:**

I have no other comments or suggestions for this paper.

**Other Strengths And Weaknesses:**

**Strengths:**
1. The proposed AGDiff method tackles the challenge of anomaly scarcity in GLAD by generating pseudo-anomalous graphs via a latent diffusion process, which surpasses existing GLAD methods and shows significant novelty and promising potential impact.
2. The theoretical analysis in appendix A well supports the validity of the proposed method, which provides empirical trends in both reconstruction errors and performance metrics to demonstrate that the generated pseudo-anomalous graphs enhance the decision boundary learning.
3. AGDiff achieves state-of-the-art performance compared to existing GLAD approaches on diverse benchmark datasets, particularly in imbalanced settings. The ablation study also validates the importance of each component. The experimental results reflects the practical effectiveness and adaptability of the proposed method.

**Weaknesses:**
1. The effect of the KL divergence term (Eq. 7) in the pre-training phase requires clarification. Can the authors further explain it?
2. Eq. 10 introduces noise $\eta$ to perturb latent embeddings, while its influence on anomaly detection performance is not discussed.
3. While the empirical results show that the generated pseudo graph anomalies improve anomaly detection, the authors are encouraged to provide some visual comparisons between the generated pseudo anomalies and normal graphs, which would improve the interpretability of the proposed method.
4. This paper refers to the “diversity” of the generated pseudo graph anomalies, but only qualitatively through the t-SNE visualization. Are there quantitative metrics, e.g., graph edit distance or MMD, to assess the discrepancy of the generated samples compared to normal graphs?

**Questions For Authors:**

Please refer to Weaknesses above.

**Relation To Broader Scientific Literature:**

Different from the diffusion model-based AD approaches [1, 2] that focus on modeling data normality, AGDiff proposes to leverage the diffusion model as the perturbator for normal data, which provides a novel and promising solution for addressing the anomaly scarcity challenge and make it distinct from supervised GLAD approaches reliant on labeled data.

[1] Diffusion models for medical anomaly detection. MICCAI, 2022

[2] Dynamic addition of noise in a diffusion model for anomaly detection. CVPR, 2024

**Theoretical Claims:**

I have checked the theoretical analysis of this paper in the appendix, where AGDiff (Algorithm $\mathcal{A}\_{2}$) is compared with reconstruction-based model (Algorithm $\mathcal{A}\_{1}$), and the related analysis is compelling.

---

> ### Author Rebuttal · Authors · 2025-04-01
>
> **We thank the reviewer for the comments. Hope our responses below are helpful in solving your concerns.**
>
> **To W1:** The KL divergence term in Eq. 7 is crucial for regularizing the latent space during pre-training. By encouraging the latent distribution $\mathbf{Z}$ to align with the standard normal prior $\mathcal{N}(\mathbf{0}, \mathbf{I})$, it prevents the model from overfitting to a narrow or degenerate latent manifold [1]. This well-structured latent space is key for generating diverse pseudo-anomalous graphs in subsequent stages. We will highlight this in our paper.
>
> ```
> [1] Kipf, Thomas N., and Max Welling. Variational Graph Auto-Encoders, NeurIPS Workshop on Bayesian Deep Learning, 2016.
> ```
>
> **To W2:** We have tested the effect of different noise magnitudes $\eta$ on MUTAG and ER\_MD, as summarized in the table below.
>
> $\begin{matrix}
> \\hline
> \text{Datasets}&\text{Metrics}&\eta=0&\eta=0.01&\eta=0.1&\eta=1&\eta=10&\eta=100\\\\\\hline
> \text{MUTAG}&\text{AUC}&93.20(2.64)&92.64(0.32)&92.40(0.08)&\bf95.83(2.15)&92.00(0.00)&92.00(0.00)\\\\
> &\text{F1-Score}&86.00(2.00)&88.00(0.00)& 88.00(0.00)&\\bf89.45(1.37)& 88.00(0.00)& 88.00(0.00)\\\\\\hline
> \text{ER\\_MD}&\text{AUC}&82.95(2.17)&86.81(0.91)&87.78(1.07)&\\bf91.21(1.84)&82.08(0.52)&81.76(0.59)\\\\
> &\text{F1-Score}&77.36(1.89)&80.19(2.83)&85.09(0.94)&\\bf86.04(2.26)&71.70(0.00)&72.64(0.94)\\\\\\hline
> \end{matrix}$
>
> We can observe a performance degradation when this noise term is removed (i.e., $\eta=0$) or set to an excessive value (e.g., $\eta=10$). Our explanation for these observations is that (1) removing the noise term may lead to over-proximity between the pseudo graph anomalies and the normal graph, thus making it difficult to train an anomaly detector. (2) While an excessively high $\eta$ can lead to pseudo anomalies that deviate too far from normal data, thus reducing the discriminative ability of the anomaly detector. These findings highlight the importance of balancing $\eta$ to ensure meaningful perturbation to the initial latent representation $\mathbf{Z}_{0}$.
>
> **To W3:** We have added a visualization comparing pseudo anomalies with normal graphs, available at https://anonymous.4open.science/r/AGDiff-137F/visualization/ (under the "visualization" file). The visualization results clearly illustrate that while the pseudo anomalies lie close to the normal data (with certain overlaps), they consistently exhibit subtle deviations, which serves as good evidence to support our claim. We will include these visual comparisons in the paper.
>
> **To W4:** First, we have performed a quantitative analysis of the diversity of pseudo-anomalous graphs on ER\_MD. Except for that, rather than graph edit distance (which has $\mathcal{O}(n^{3})$ complexity), we employed the **normalized Laplacian spectral distance** to measure the discrepancy between the generated anomalies and normal graphs, as it is computationally efficient and direct related to graph topology. Specifically, we computed pair-wise spectral discrepancies between:
> 1. **Generated pseudo-anomalous graphs and normal graphs**, which ranged in $[0.17, 1.12]$.
> 2. **Real anomalous graphs and normal graphs**, which ranged in $[0.40, 5.12]$.
>
> While the range for pseudo anomalies is narrower compared to real anomalies, this outcome aligns with our design intent. Rather than mimicking the extreme deviations observed in real anomalies, we aimed to produce controlled yet challenging perturbations to enhance the decision boundary learning. The embedding visualization results (in our response to W3) further support this, where the pseudo anomalies are distinctly separable from normal graphs. These results indicate that the generated pseudo anomalies exhibit a meaningful discrepancy from the normal graphs, providing sufficient diversity to effectively challenge the model during training.

---

> > ### Comment · Reviewer_kMMx · 2025-04-02
> >
> > I appreciate the authors’ detailed responses, which adequately address my concerns, particularly regarding the interpretability issue. The additional visualizations and statistical results further justify the motivation and the validity the proposed method. In light of these improvements, I would like to raise my overall assessment to this work.

---

> > > ### Author Response · Authors · 2025-04-03
> > >
> > > **Dear Reviewer kMMx:**
> > >
> > > We appreciate your time and effort in revisiting our work, your positive comments mean a lot to us. We are pleased that our responses can help address your concerns.
> > >
> > > Best Regards,
> > >
> > > Authors

---

### Official Review · Reviewer_PZBE · 2025-03-14

**Overall Recommendation:** 3

**Summary:**

The paper introduces a graph-level anomaly detection method to improve the performance of GNNs on anomalous graphs. The method consists in generating pseudo-anomalous graph with diffusion models in other to enhance the classification capabilities.

**Claims And Evidence:**

The paper claims to that generating pseudo-anamalous graphs improves the learning capabilities of GNNs.

**Essential References Not Discussed:**

See item before.

**Experimental Designs Or Analyses:**

I believe the experimental design does not help in answering the questions.

**Methods And Evaluation Criteria:**

The paper fails to verify the claim that it puts forward.

1. What are pseudo-anomalous graphs? Why are they pseudo-anomalous and not simply anomalous?
2. This method is simply data augmentation utilizing a generative model. What is the novelty of the method beyond putting two existing pieces together?
3. There is no proof that the improvement of the method is actually because of the pseudo-anomalous data. By are the authors claiming this? How do they verify this?

**Other Comments Or Suggestions:**

None.

**Other Strengths And Weaknesses:**

I believe the paper has very little novelty.

**Questions For Authors:**

See before.

**Relation To Broader Scientific Literature:**

I believe the paper is a data augmentation paper.

Data Augmentation for Supervised Graph Outlier Detection via Latent Diffusion Models
Kay Liu, Hengrui Zhang, Ziqing Hu, Fangxin Wang, Philip S. Yu

Counterfactual Data Augmentation With Denoising
Diffusion for Graph Anomaly Detection
Xiao et al.

**Theoretical Claims:**

The authors do not put any theoretical claims forward.

---

> ### Author Rebuttal · Authors · 2025-04-01
>
> **We thank the reviewer for the comments. Hope our responses below help to solve your concerns.**
>
> **To Q1:**
> 1. **What are pseudo-anomalous graphs?** Pseudo-anomalous graphs are graphs generated via a controlled latent diffusion process (refer to Section 4.3). These graphs are optimized to resemble normal graphs yet exhibit subtle deviations, therefore mimicking potential anomalies and providing effective auxiliary signals for refining the decision boundary.
> 2. **Why are they pseudo-anomalous and not simply anomalous?** The graphs are termed "pseudo-anomalous" as they are not sampled from real anomaly data, but are generated by introducing subtle perturbations to normal graphs. In unsupervised GLAD, real anomalous data is not utilized during training. Therefore, our approach leverages pseudo-anomalous graphs to compensate for this absence.
>
> **To Q2:** Please note that our work focuses on GLAD, and is not merely a data augmentation paper. We want to highlight two key innovations of AGDiff beyond standard augmentation:
> 1. AGDiff leverages a conditioned latent diffusion process to generate pseudo-anomalous graphs from normal data (Section 4.3), which is not arbitrary augmentation but a targeted approach to simulate potential graph-level anomalies via controlled perturbation in an unsupervised manner, which addresses the anomaly scarcity problem in GLAD. Therefore, our approach is distinctly different from GOEM [Liu et al.]’s supervised diffusion process (relying on labeled anomalies) and CAGAD [Xiao et al.]’s counterfactual augmentation (based on modifying the neighboring node attributes).
> 2. Unlike the traditional augmentation paradigm where augmentation is decoupled from learning, AGDiff integrates the diffusion process and anomaly detection in a joint training framework (Section 4.4). This enables bidirectional feedback where the detector refines the generation process, and the generator is adapted based on feedback from the detector to generate more challenging pseudo-anomaly samples.
>
> Moreover, we want to emphasize that GOEM [Liu et al.] and CAGAD [Xiao et al.] are designed for node-level tasks, which differs fundamentally from AGDiff. Our approach is specifically tailored for graph-level anomaly detection, addressing the scarcity of anomalous data with an unsupervised and adaptive strategy.
>
> **To Q3:** The positive effect of the pseudo-anomalous data can be validated in the following parts of our paper: (1) In the ablation study (Table 3), the performance significantly drops ($\geq$ 20\%) when we remove the latent diffusion module (no pseudo-anomalous data generated) or the conditioned vector (determine the generation quality of pseudo-anomalous data). (2) Refer to Appendix A - through the theoretical analysis and empirical comparison with the reconstruction-based model, we demonstrated the benefits of pseudo-anomalous data in improving the discrimination capability of the anomaly detector. These findings are good evidence of the contribution of pseudo-anomalous data.
>
> **To Q: Theoretical Claims:** We theoretically and empirically demonstrated why AGDiff could outperform the reconstructor-based approach. (Refer to comments of Reviewer kMMx, BUES, and 2Zh2)
>
> **To Q: Experimental Designs Or Analyses:** Here are the summarized questions that our experiments answered:
> ```
> 1. How does AGDiff compare with SOTA GLAD baselines? (Refer to Table 1)
> 2. How does AGDiff perform in large-scale imbalanced GLAD? (Refer to Table 2)
> 3. How can AGDiff's discrimination ability be verified? (Refer to Figures 2 & 5)
> 4. How does the hyper-parameter impact the performance? (Refer to Figure 3)
> 5. How does each component in AGDiff contribute to performance? (Refer to Table 3)
> 6. Why can AGDiff outperform reconstruction-based methods? (Refer to Appendix A)
> ```
> We have further answered several questions in the rebuttal (see responses to Reviewers kMMx's W2\&W4, BUES's W2\&W4, 2Zh2's W3). If you have other questions, please feel free to discuss them with us.
>
> **To Q: Supplementary Material:**
> 1. We set the number of generated graphs equal to the normal graphs as this balanced configuration leads to more stable training. As per your suggestion, we further conducted an ablation study by varying the ratio $r$ of generated pseudo graphs relative to normal graphs. The results (on DD) below indicate that the balanced setting (i.e., 100\% ratio) yields the best performance.
>
> $\begin{matrix}
> \\hline
> &\text{Metrics}&r=30\\%&r=60\\%&r=90\\%&r=100\\%\\\\\hline
> \text{DD}&\text{AUC}&85.06(3.42)&86.72(0.70)&87.30(0.49)&\bf88.23(0.67)\\\\
> &\text{F1-Score}&80.78(4.48)&83.23(1.45)&84.03(0.36)&\bf84.06(0.59)\\\\\hline
> \end{matrix}$
>
> 2. It is important to clarify that we are not training the GNN using the diffusion model. The diffusion model is jointly trained with the anomaly detector (GNN as the backbone), and functions as a perturbator to generate pseudo-anomalous graphs. We have analyzed AGDiff's time complexity (refer to Reviewer BUES's W2).

---

### Decision · Program_Chairs · 2025-05-01

**Decision:**

Accept (spotlight poster)

**Comment:**

The work introduces a novel method for Graph-Level Anomaly Detection. Strong empirical justification on the effectiveness of the method and its design is presented.

The author did a great rebuttal that convinced two reviewers to increase their recommendation scores. After rebuttal, all four reviewers agree that the paper is above the acceptance bar of ICML.